# Making Room for Our Forthcoming Rivers

Andrea Gianni Cristoforo Nardini

Fundación CREACUA, Calle 1A n.1-109, Riohacha, La Guajira 440001, Colombia; nardiniok@gmail.com

**Abstract:** This paper provides a schematic, conceptual trip across a set of paradigms that can be adopted to design flood control actions and the associated *river setting*, including the space allocated to the river. By building on such paradigms, it eventually delineates an integrated approach to identify a socially desirable river setting, under a climate changing reality. The key point addressed is that when residual Risk and Operation, Management and Replacement costs are considered to their full extent, even a basic economic analysis may suggest alternative river settings that can be more attractive, particularly if accompanied by suitable economic-administrative management measures. Emphasis is put on the deep uncertainty characterizing the whole decision problem and on the need for a drastic change of paradigm. The approach proposed can greatly improve current Flood Risk Management Plans responding to the European Flood Directive (Directive 2007/60/EC). It can also help to develop constructive dialogues with stakeholders, while enhancing the understanding of the problem. Although mainly intended to address a conceptual level, it also aims at providing an applicable method.

**Keywords:** flood risk; fluvial space; river management; climate change; paradigm





## 1. Introduction

It is probable that all of us are progressively interiorizing, slowly or quickly, the climate change issue. A spontaneous question arises: "new climate → new river?" I mean by this expression an indeterminate, intuitive idea that our rivers will somehow be modified as a consequence of a new climate (and the associated different hydrological regimes, sediment supplies and transport, and vegetation dynamics).

A lot of ongoing research relates to this issue mainly focusing on the recent modifications experienced by water regimes (e.g., [1]), with longer or shifted drought periods (e.g., [2]), and more frequent and larger scale flood events (e.g., [3]), and even by riparian vegetation (e.g., [4,5]), or biota (e.g., [6,7]). Research is growing to speculate about the future evolution of such aspects, including the probable geomorphological changes of rivers [8–10], starting from a new planform and, in simple words, the associated *fluvial space* required. A river may display a narrower bed because of vegetation encroaching due to longer dry periods and lower rejuvenation effects [11]. On the contrary, more frequent intense precipitation events, and an associated change in the extent and type of vegetation cover of the catchment, may increase the expected solid transport capacity and broaden the bankfull channel. In any case, expected harsher, and more frequent, flood events, jointly with higher sea levels, will involve a wider territory and most probably will require a change of the *river setting*, a locution that herein means the *space allocated to the river, with the land use in it, together with the set of hydraulic works aiming to control fluvial dynamics and flooding or to exploit water resources*. A different river setting implies, of course, a corresponding different planform, morphological configuration and geomorphic/flooding dynamics.

The main thesis of this paper is that a rethinking of river settings is unavoidable and that this issue should strongly permeate the elaboration of the Flood Risk Management Plans required by the Flood Directive (FD) (Directive 2007/60/EC on the assessment and management of flood risks); furthermore, it would also lead to a redesign of the rural and

urban territory. The socio-political-economic difficulties involved in such a demanding change are undoubtedly extremely large, while an enormous dose of courage is required.

This paper explores schematically several paradigms and associated approaches that can be adopted to design flood control actions and the associated river setting. On this basis, a conceptual-technical framework and methodology that can support the identification of a socially desirable river setting, and its associated fluvial space, under a climate changing reality, is progressively delineated. Other related issues, such as water quality, water regime and biodiversity are not touched, although they certainly deserve similar considerations and are strongly related.

This is a concept paper aiming at stimulating reflection and debate. It does not present case studies nor an exhaustive review of research papers (a selection of papers relevant to support the arguments developed is adopted), but it involves evidence easily verifiable by everyone. Most of what is presented here is well known; the claimed novelty is in presenting key concepts in a new way, while linking them together in a structured and robust discourse, and offering a pathway for an improved, clearly stated approach.

The paper starts with a very synthetic presentation of what can be denominated as the *classic engineering paradigm* of flood control, pointing out its weaknesses (a point of view shared by many others, for instance, [12] or [13]). It is then shown how more evolved paradigms spontaneously derive from it, leading to a Cost–Benefit Analysis paradigm (CBA). Climate change is then considered and arguments provided to conclude that an important paradigm shift is accordingly required; its backbone is to restitute space to the river by defining an appropriate river setting within a wide corridor. The role of the CBA as a tool to define and assess candidate river setting alternatives at a preliminary level is illustrated. The focus then switches to uncertainty by pointing out, on the one hand, that a strict uncertainty approach is appropriate because of our ignorance about future probability distributions of hydrological variables; and, on the other hand, by delineating an operational approach to dealing with hydrological variability within any future climate scenario. A further insight is developed by addressing the idea of adaptive planning, in the face of an uncertain future, so clarifying the meaning of *flexibility*. Finally, it is stressed that even the extended version of the CBA cannot be considered the panacea; a brief discussion is then provided about the need to broaden the view to directly address people's quality of life—of which risk is just one of the relevant components—so achieving a further evolved paradigm.

## 2. Evolving Paradigms Guiding River Setting

The present situation of rivers is, in general, very unsatisfactory from several points of view, as can be easily ascertained by the direct experience of:

- ecosystem health;
- flood risk (and damages);
- management costs.

Health (the ecosystem status) of European rivers is even worse than reported by the EU Member States according to the Water Framework Directive (WFD) requirements (Directive 60/2000/EC establishing a framework for community action in the field of water policy), and available on the WISE website (https://www.eea.europa.eu/data-and-maps/data/wise-wfd-4, accessed on 27 October 2021). In fact, the reported indices are in general limited to water quality and water biota, while the morphological component comes into play only for water bodies with such components in the elevated category [14] or amongst the pressures. Most of the rivers see their morphology and riparian vegetation disrupted by uncountable exploitation and "river training" engineering works which together ensure a very low ecosystem health (this has motivated, amongst others, the AMBER project and its initiative to map all types of barriers: https://amber.international/portfolio-types/barrier-removal/, accessed on 27 October 2021).

To confirm that flood risk is very high it is sufficient to look at recent events in several countries, amongst which, for instance, Germany (event of 12–15 July 2021: https://www.

worldweatherattribution.org/heavy-rainfall-which-led-to-severe-flooding-in-western-europe-made-more-likely-by-climate-change/, accessed on 25 October 2021) and Italy (several events during October 2021, particularly in Sicily).

Evidence that the costs related to investment in, and maintenance of, control works are so high that society, in fact, often does not cope with them in due form can easily be found by observing the number of works in a poor state along many rivers, or the fact that a large portion of the management actions planned so far have not been implemented because of a lack of funds.

### 2.1. Classic Engineering Paradigm

This situation is a consequence of the hydraulic engineering paradigm that can be schematically represented by the motto "control the river and put the territory in safe conditions", i.e., "achieve safety". It is detailed by the following criteria (here the symbol "→" means "this leads to implement"):

- keep river channels hydraulically efficient so as to let the flood flow as quick as possible → dredging, recalibration, de-vegetation, rectification;
- keep the water within the channel, avoiding overflows → levees, by-pass;
- keep the river (and the terrain slopes) in place, avoiding any movement → bank defenses (rip rap, dikes, groynes), weirs, torrent control works, etc.

It is accompanied by a cost-effective planning approach that can be formalized as shown in Figure 1 and that, in summary, says (the **bold** style is adopted herein to denote vectors):

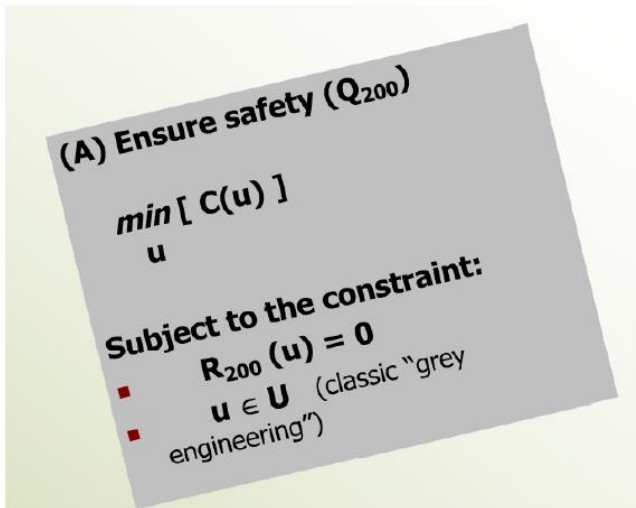
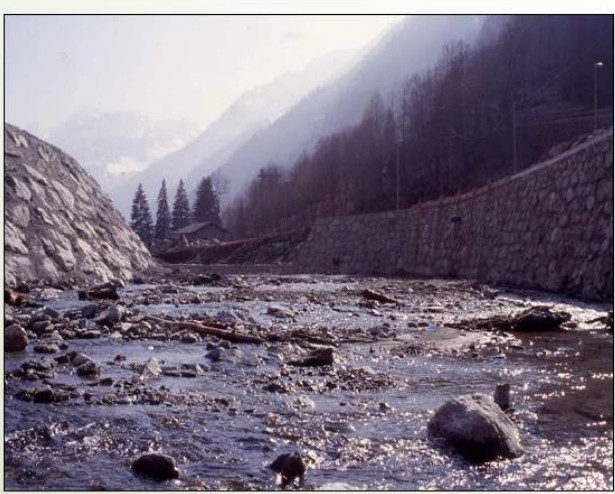

**Figure 1.** The classic hydraulic engineering paradigm: (**left**) schematic formulation; (**right**) an emblematic example of intervention (Photo by A. Goltara, T. Noce, Northern Italy).

*ensure that, given a reference event (e.g., a 200-year return period flowrate $Q_{200}$), the associated flood risk $R_{200}(\mathbf{u})$ is nullified (i.e., all lower events will produce no damages), while trying to spend an amount $C(\mathbf{u})$ as little as possible, and choosing interventions ($\mathbf{u}$) mainly aiming at reducing the hazard (rather than the exposed value), from a set $\mathbf{U}$ of classical "grey" ones (levees, dikes, reservoirs, recalibration of river channels, dredging, by-passes, etc.).*

This paradigm is so deeply embedded in the collective consciousness that the output from a hypothetical large-scale enquiry would very probably be that a large majority of people see the flood risk problem as follows:

- *Damages occur because rivers are not "clean": there is need to dredge sediments and eliminate riparian vegetation* (notice that this misuse of the term "clean" is a reality in newspapers and in the general language of non-technical policy makers);

- *There is not enough money for the defense works that would save us: there is need for political pressure.*

All this spontaneously leads to the strong conviction that, if rivers were cleaned and defense works built (i.e., engineering structures aimed at controlling the river behavior), the flood problem would be overcome. As a result, in many cases people have just forgotten (or ignored) the presence of a nearby threatening water course, as demonstrated by the development of urbanized areas that are often shoulder-to shoulder with the water course or even have buried it underground.

Unfortunately, the "safety" ideal inspiring the classical approach is a mere chimera, because:

- *Schemes based on defense works are likely to often respond not as designed (e.g., multi-purpose reservoirs will not be empty at the critical moment because the event may occur in an unexpected period; a retention tank will be filled by a relatively small pre-flood and then ineffective for the successive large one; an overflow canal may be obstructed by a small local landslide, or the modeling exercise carried out in the prediction was imprecise or even incorrect);*
- *Events exceeding the design threshold are always possible (even without invoking climate change);*
- *The river is not only water, there is a geomorphological dynamic that can significantly change its behavior in time because of processes such as incision, aggradation, meandering, avulsion, etc.;*
- *A system controlled by artificial structures is (extremely) fragile: such structures may fail (even for events lower than the design one) and the higher their presence, the more likely the failure (see Figure 2, left);*
- *People think they are protected by defense works and hence safe (Fekete and Sandholz, [15] discuss this for German society); in addition, exposed value typically increases and risks increase with it (see Figure 2, right) (regulations are often not accomplished).*

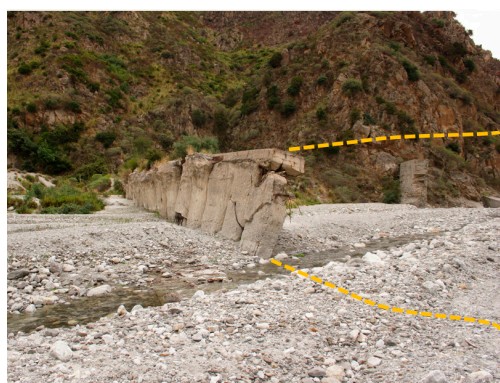 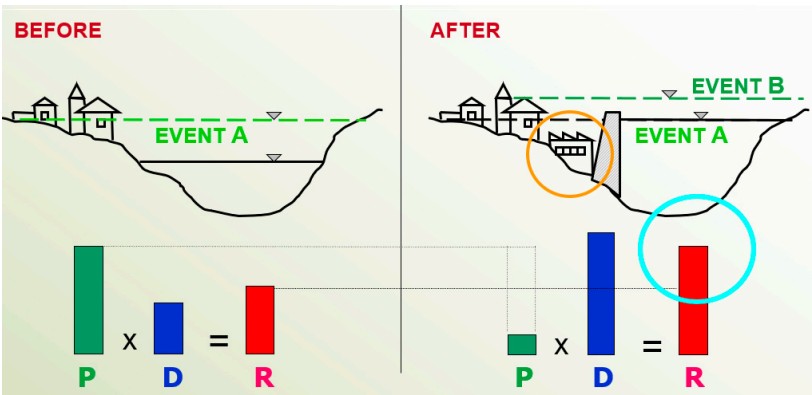

**Figure 2.** (**Left**): Evidence of two failed big weirs originally conceived to control downstream aggradation: a stout, but fragile, solution; (**Right**): The paradox of putting a system in safe conditions leading to an increase in risk with respect to the original situation (BEFORE sketch) because people, feeling safe after the protection works (the longitudinal dike in the AFTER sketch), inevitably increase exposed value (golden circle and higher blue D bar of potential damage; not in scale); but an event higher than the design one (Event B higher than Event A) is perfectly possible, although with a lower occurrence probability (P); Risk (R), a combination of probability P and damage D, has hence grown.

Moreover, "forgetting the river" deletes the awareness of the risk problem as well as that of a socially, environmentally and economically very valuable asset.

### 2.2. Total Risk Paradigm

The real and serious possibility of events overcoming the defense system, because of flows higher than the design flow, and of the possible collapse of some works—i.e., the so-called *residual risk*—pushes us towards a more comprehensive approach where the

residual risk $R_R$ is incorporated within the total risk $R_T$ now to be minimized, as depicted in Figure 3, i.e.:

$$R_T = R + R_R \tag{1}$$

where R is the risk within the territory (partially) protected by defense works assumed to work properly and under events up to the reference one (adopted for the design). This is generally an almost zero figure (according to the classic engineering paradigm); but in practice there often still is a (low or high) risk as the whole territory is not protected and some uses may still be allowed in it.

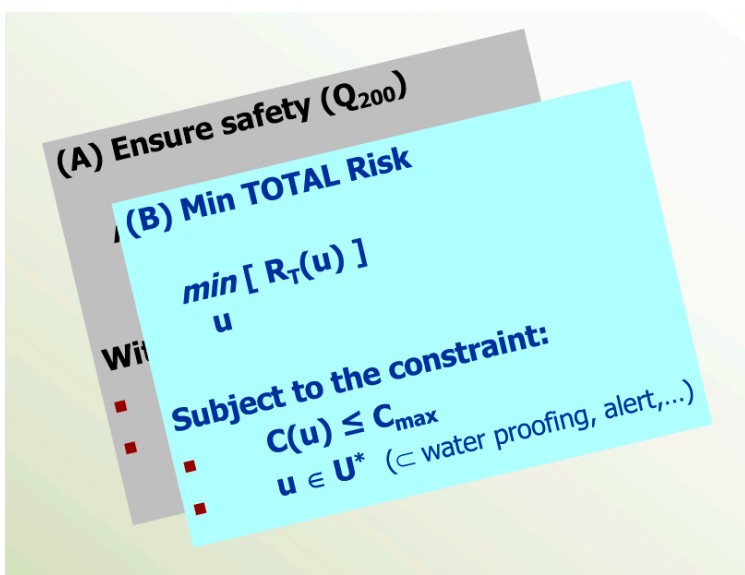

**Figure 3.** An evolved paradigm centered on the total risk $R_T$ including the residual risk ($R_R$), and a broader spectrum **U\*** of possible interventions, under a maximum expenditure constraint $C_{max}$. A further evolution may recognize a multidimensional nature of the risk (e.g., tangible, intangible).

This is indeed a change of paradigm because we now admit that risk cannot be eliminated, but just reduced, and that the standard risk component R may be traded off by the residual risk component $R_R$, while this, in turn, requires much more attention than that allocated up to now (as stressed, for instance, by Wagner et al. for West Africa [16]), capitalizing current and future research (e.g., the Rescue project, https://cordis.europa.eu/project/id/625258/reporting/it, accessed on 27 October 2021). The conviction that risk cannot be eliminated, but just reduced, belongs to the common practice of all engineering projects. However, when referring to flood risk, it is in fact a new achievement, as demonstrated by the increasing attention of recent studies on possible avulsions in important water systems (e.g., [17]).

It is assumed here that the probability of events—including very infrequent ones—can be reliably determined in order to compute $R_R$. This may not be the case, according, for instance, to the strong arguments of [18], such as irremediable shortness of required data records (at least 10 times longer than the considered return period) or non-stationarity of underlying physical processes (e.g., floods generated by Glacial Lakes Outburst Floods). The reasoning which follows here nevertheless still holds, at the cost of waiving the compactness of CBA (i.e., a judgment vehiculated by one main indicator, the net benefit) and switching to a multi criteria approach with a qualitative, but still meaningful, measure of $R_R$.

The set of possible interventions (options) is broadened by including those aiming at reducing the vulnerability of the exposed assets—particularly outside the corridor, especially in the urban territory, such as *water proofing* (e.g., [19,20]) and alert systems. Nature-Based Solutions (e.g., [21]), such as dismantling of some protections to recover flooding space (experiences are available also for Mediterranean countries such as Spain,

e.g., in the Duero basin, https://www.chduero.es/web/guest/estrategia-nacional-de-restauracion-de-rios, accessed on 27 October 2021) and Spatial Planning, also fit here, as increasing flooding in rural areas may decrease risk in other, more valuable (urban) areas. A fascinating history of the evolution of the structure of the flood control decision set is offered, for instance, by Lonnquest et al. [22] by comparing the experiences of the Dutch and the Americans.

### 2.3. The Cost-Benefit Analysis Paradigm

The residual risk is intimately linked to a quite unpleasant, but unavoidable, guest of the river management arena: the *Operation, Maintenance and periodic Replacement* cost of all works (OMR). As soon as a new work of any type is born, it brings to life the collateral curse of the never-ending obligation to take care of it, by operating it, making some repairs when needed, and sooner or later replacing it in full. Unfortunately, this term is too often disregarded or heavily underestimated in new projects' evaluation.

It can be given a formal shape as follows:

$$C_{OMR}(\mathbf{u}) = \sum_{t=0,1,2,\dots T} \delta^t \left[c_{OM}(\mathbf{u})\right] + \sum_{\tau=0,1,\dots,\mathrm{int}(T/\Delta)} \delta^{(\tau*\Delta)} \left[c_C(\mathbf{u})\right] \tag{2}$$

where: $C_{OMR}(\mathbf{u})$ = total, present day OMR cost, over the planning horizon, depending on the vector $\mathbf{u}$ of decision options, i.e., the interventions put in place (in the following, for simplicity, $C_{OMR}(\mathbf{u})$ is denoted simply as OMR); $c_{OM}(\mathbf{u})$: operation and management annual cost, depending on the vector $\mathbf{u}$, occurring every year on average; $c_C(\mathbf{u})$: construction cost, depending on the vector $\mathbf{u}$. This occurs every $\Delta$ years, because of reconstruction, being $\Delta$ the average lifetime of works (rigorously speaking, it should be differentiated on the typology of intervention), with:

- $\delta = 1/(1 + r)$: discount factor with the social interest rate r;
- T: planning time horizon (years).

Its link with the residual risk $R_R$ is made evident by the example shown in Figure 4; that is, the lack of due OMR expenditures consumes the "work-in-good-shape" capital, opening the door to a work malfunctioning or its collapse.

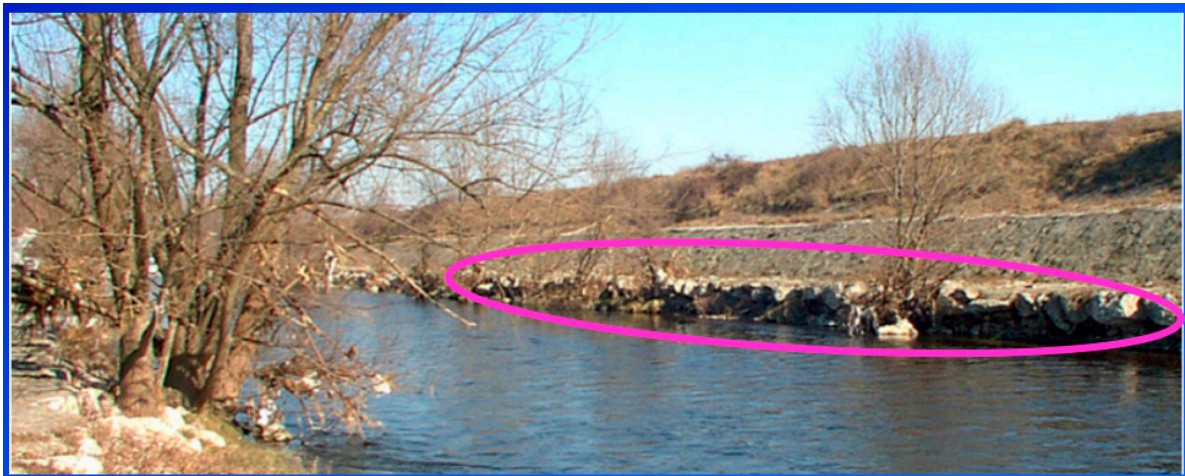

**Figure 4.** An example of consumption of the "work-in-good-shape" capital, because of lack of maintenance and periodic replacement (river Chiese, Italy): the toe of the bank protection is disrupted (fuchsia circle) and the door of a defense collapse is ajar.

This reality pushes us to admit that our planning exercise has to consider a balance between benefits and costs. This is true in general. In the river management domain, a trade-off can be sought between the risk, on the one side, and, on the other side, the cost of

measures aimed at reducing it, particularly if the same community has to pay for (a good part of) them, a suggestion also pointed out, for instance, by [23].

This, again, is a shift of paradigm, where society now looks for the best allocation of resources and finds in the Cost Benefit Analysis its natural planning tool (CBA, e.g., [24–27]; a recent, emblematic case study in USA is presented in [28]). This is depicted in Figure 5 in its basic form, where just total risk and total costs (including investment I and OMR) are considered. All terms are intended as differentials with respect to a reference situation which implies assessing all figures twice, as described for risk, for example, in [29]). Environmental services (other than flood risk reduction, for example, carbon capture or recreation enhancement) can and should be incorporated (e.g., [30–35]). For the moment, however, they are left outside because their quantification is generally delicate (difficult, unreliable and questionable), while costs and risk are more solid (at least for what concerns the direct, tangible risk component for which assessment techniques exist, such as [36–39], although continuously evolving). Furthermore, the costs of damages and works involve real money, while the former component (when touching no market factors) often involves hypothetical money, that can generally take a concrete consistency only if (complex and delicate) payment schemes are put in place. Other components can anyway be added later on; first of all, the externalities on water usages (i.e., a loss of environmental service) due to the impact on exploitation works associated with a new river setting, but also the increase of economic value of residential assets because of higher safety.

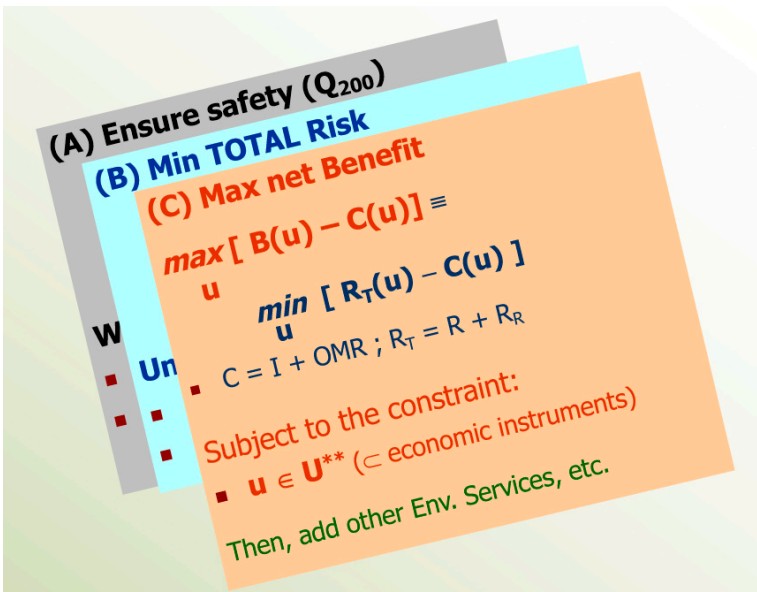

**Figure 5.** Costs count too: a basic CBA paradigm (B stands for Benefits and C for Costs) including the total risk $R_T$ and the total cost C (investment I plus OMR), all depending on planning decision **u** belonging to a wider set **U\*\***. The terms are differential with respect to a reference alternative, e.g., present situation where I is null.

The risk component involving intangibles as, in particular, human life can still be measured economically (e.g. by questionnaires assessing the willingness to pay or to accept; or as the capitalized value of the wages during the expected working life), but such methods are hardly acceptable, particularly in the EU context. A sound alternative, is to include a new constraint (not represented in the Figure 5) as *the number of likely deaths for all events with very large probability of not being exceeded be null*.

The set of possible interventions is further widened by including nonstructural measures, i.e., economic-administrative-sociocultural instruments such as indemnification mechanisms, insurances, Payment for Ecosystem Services (PES), further specific agreements (e.g., [40]), or information and education campaigns.

## 3. The Overflowing Quest of Climate Change for a More Daring Approach

The approaches presented above do not only satisfy a different and evolving conceptual understanding of the problem; they may also lead to very different solutions.

Notice that no climate change is invoked yet in the above discussion!

Climate change is a reality. This statement is gathering more and more consensus. The recent IPCC AR6 report [41,42] presents more frightening findings and predictions than before. The empirical, independent evidence, for instance from the European Severe Weather Database (ESWD), seems to confirm this trend (Table 1), although increasing urbanization and land consumption, river artificialization and improvement of recording protocols may be, of course, concomitant causes of this evidence. Recent events, such as that in Germany/Belgium (14–15 July 2021) with harsh damages in an area not considered at particular risk [15], or that in Rossiglione (Liguria, Northern Italy), with an extremely intense rain event with approx. 740 mm in 12 h and 880 mm/24 h or more on 7 October 2021 (according to Centro Meteo Ligure di Genova, Italy, https://www.centrometeoligure.com/, accessed on 11 October 2021), are expected to soon lose the connotation of statistical "outliers" and rather become an unexpected new "normality".

**Table 1.** An example showing that the frequency of harsh hydrological events (here just "heavy rains" and "avalanches") is in general growing (https://www.eswd.eu/, accessed on 11 October 2021).

| Year | Italy | Spain | France |
|---|---|---|---|
| 2010 | 194 | 44 | 57 |
| 2020 | 314 | 243 | 198 |
| *increase: %* | 62 | 452 | 247 |

### 3.1. Climate Change: Just a Matter of Modifying Return Periods $T_R$?

Technically, to deal with a different climate, it may seem sufficient to just utilize higher flowrates $Q^*(T_R)$ associated with each return period $T_R$ (rather than those $Q(T_R)$ obtained by statistical analysis of historical time series), and then proceed with the usual approach.

However, this is a largely over-optimistic assumption, first of all because the uncertainty associated with future climate is enormous, as also stressed by the recent IPCC AF6 report [42], indeed:

- Global Circulation Models (GCMs) provide today very different answers in terms of average precipitation ($\bar{p}$) and temperature ($\bar{t}$) (Figure 6); even more significant is the difference in terms of climate variability [43] which, eventually, is the key feature driving flood risk and is predicted to increase by more than average figures (e.g., [44]);
- A changing climate may not only increase the magnitude of an event for a given $T_R$, but even modify the climatic mechanisms responsible for the formation of floods. In such a way, a large basin may see events never registered before, where possibly several significant tributaries may experience a flood simultaneously. An example was experienced in 2006 by the Danube when an extremely rare coincidence of relatively large floods occurred in the subbasins of the Upper Danube at the same time as flooding on the Tisza, Sava and Velika Morava, and led to a very serious 100-year flood event along more than 1000 km of the river [45];
- Scientists are making big efforts to predict climate by GCMs, downscaling, fascinating, complex machine learning techniques and others (e.g., [46]). However, possibly the largest unknown factor is how nations will indeed behave in terms of carbon emissions reduction (or increase); or—very sad to say—what will be the effect of the current war in Ukraine: more emissions because of a rebooted use of coal, or less because of an accelerated development of alternative energy sources? Last, but not least, is the uncontrollable contribution by terrestrial, and particularly subaerial volcanos, or permafrost melting, amongst others.

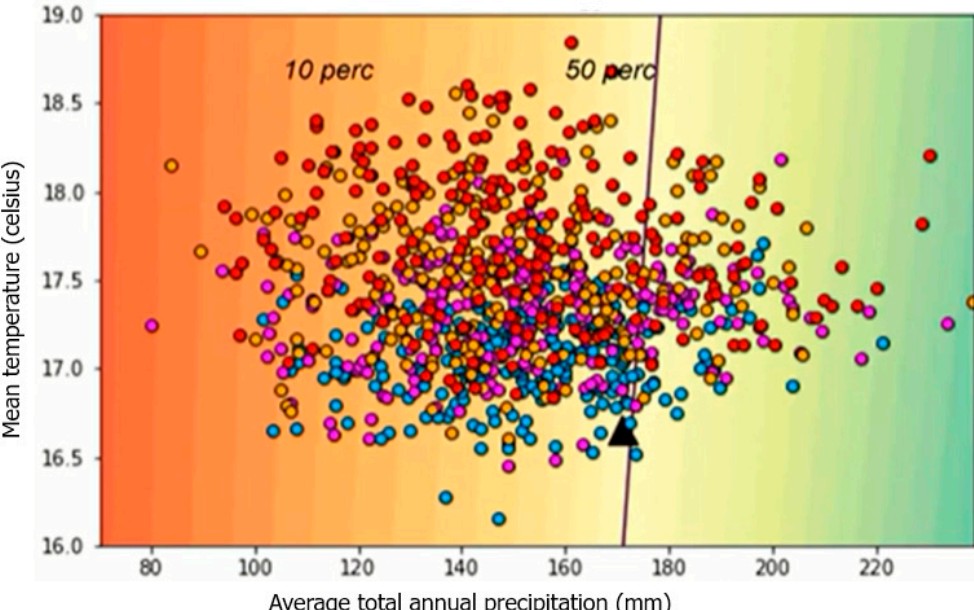

**Figure 6.** Example of dispersion of predictions of GCMs in terms of annual average precipitation (p) and temperature (t) (adapted with kind permission from [47]). Here the outputs are temporally averaged projections of the SIMGEN weather generator; the forced anthropogenic trend component is inferred from the 50th percentile trend that is projected by GCMs within the CMIP5; dots' colors correspond to averages along different future periods (blue: 2020–2030; pink: 2030–2040; orange: 2040–2050; red: 2050–2060). Notice that dispersion is very pronounced and that changes with respect to present climate (black triangle) may even result beneficially (to the right of the oblique line marking a threshold of users' satisfaction). Climatic variability, in addition to the change of average values here represented, opens an additional, broader spectrum of uncertainty (e.g., [48,49] showing that its change is more significant than that of average values).

This is not at all to reduce the importance of such efforts, but to stress that uncertainty cannot be tamed to the level we would like and need.

An agreed-upon statement is that events will become harsher and harsher, while sea levels (a hydraulic boundary condition) will keep rising, reaching, in a not too far future, frightening heights. It is probably not irrelevant to note that IPCC AF6 does not exclude increments of the order of 15 m by 2300! (p. 29) [41].

The technical oversimplification mentioned above finds another pillar in the fact that the return period ($T_R$) concept, so beloved by hydraulic engineers (as confirmed for instance by [50]), conceptually is losing its meaning and should no longer be utilized because it is applicable to a cyclo-stationary climate which, by definition, is negated by climate change (as already noticed for instance in [51]). We need to recognize that, while before we knew uncertainty, as we could characterize it by probability distributions (and $T_R$), we are literally ignorant about the future, as any probability estimate would be definitely weaker than the estimates we have used in the past. This is an unpleasant *unkunk* in our hands ('unkunk' being a label which the United States Air Force was using for unpredictable problems, or unknown unknowns, see [52]; accordingly, 'kunk' can be defined as a 'known unknown'). In the words of Klemeš, (p. 10) [53] we therefore get a respectable warning about how to deal with future uncertainty: " . . . *unkunks represented as kunks become skunks*.", because, as is well known, skunk is a known which stinks.

More importantly, adverse consequences are likely to reach literally unbearable levels. Indeed, events will be harsher, more defense works will be involved (and will fail) and exposed value is growing as a consequence of the anthropization of the territory. Therefore, if a constant level of risk is pursued, then the total cost C will have to be much higher (more defenses, adaptation or land use change will be needed); if, vice versa, we want to keep costs at the same current level, a much higher risk will have to be accepted as residual risk $R_R$ will inevitably grow enormously.

This perspective says that going on with the current approach will lead to socially, financially and environmentally unbearable situations.

A modified water regime and sediment supply (because of modified climate, land cover and soil conditions, as witnessed for instance by the very recent rock fall in the Dolomites: https://blogs.agu.org/landslideblog/2021/10/12/a-large-rock-slope-collapse-from-punta-dei-ross-croda-marcora-in-the-italian-dolomites/, accessed on 14 March 2022) will also drive geomorphological changes in our river channels and their behavior, involving additional widespread complications. The importance of this aspect has been already noted in relation to the EU Flood Risk management Plans assessment, e.g., by [14].

Finally, how can we dare to pronounce "sustainability" if we keep managing rivers the same way and so will leave to our future generations an unbearable burden of works to maintain (and rebuild), which will not produce the claimed safety for the reasons illustrated above?

Dealing seriously with Climate Change cannot be reduced to the oversimplified approach depicted above. A substantial change of paradigm in flood management is needed.

### 3.2. Time Horizon

The considerations on climate change may sound too much Cassandra's screams than what is really needed. Indeed, when the IPCC calls into play far dates such as 2100 or even 2300, most laymen, and even several decision makers, and myself, will inevitably see them as somehow exaggeratedly far in the future. However, this is a problem of perception. To see it, just consider that given a person A today, A's nephew will probably see 2150! Indeed, if A has a son B today, B will probably have his own son C in his 40s, and this son C is expected to live possibly 90 years or so (we refer here to an industrialized country, for instance Italy).

Such a horizon is therefore by far a more appropriate one to deal with our problem of river setting, than the ridiculous $15 \div 30$ years horizon already considered too daring in land use and urban planning exercises. Consequently, current EU Flood Risk Management Plans cannot see a sufficiently far future to make decisions really fitting the climate change challenge, for their decisions might and should significantly change land use and urban setting starting from now.

### 3.3. A More Daring Paradigm in a Systems View

A starting point is a very basic consideration: *present rivers' setting will no longer be able to bear a new, harsher climate*. This implies that rivers (and water courses in general) will need more space, i.e., wider *river corridors*.

Another point is that it would be at least infantile to assume that defense works are able to avoid flooding into the urban territory, i.e., cities. This means that we need to broaden our attention to cover not only the river corridor, but also what lies outside of it, particularly the urban areas (Figure 7). This is to say that cities must be equipped and prepared to live with (hopefully infrequent, certainly undesired) floods.

Or, in other words, the implicit assumption of the classic engineering approach "you are safe → forget to worry about the presence of the river" has to be strongly rejected.

Uncertainty about the future is enormous; no prediction effort can get rid of it and is not even able to provide reliable probabilities. We actually face a difficult situation with an uncertainty of several natures (mainly *aleatory*, but also *epistemic*—because of incomplete and imprecise knowledge of the world—and even from *ambiguity*, due to

different interpretations of the global warming issue), characterized by a very high level, actually *ignorance* (according to the classification proposed by Jens Christian Refsgaard's IAHR seminar of 14 October 2021: https://www.iahr.org/en/lives/details?live_id=89&video_id=847, accessed on 21 October 2021; also see [54]). With strong uncertainty and no knowledge of probabilities, a consistent decision approach should now be based on a *strict uncertainty* framework [55], where the most intuitive and perhaps most sensible criterion is *risk aversion*, a cousin of the EC's precautionary principle [56].

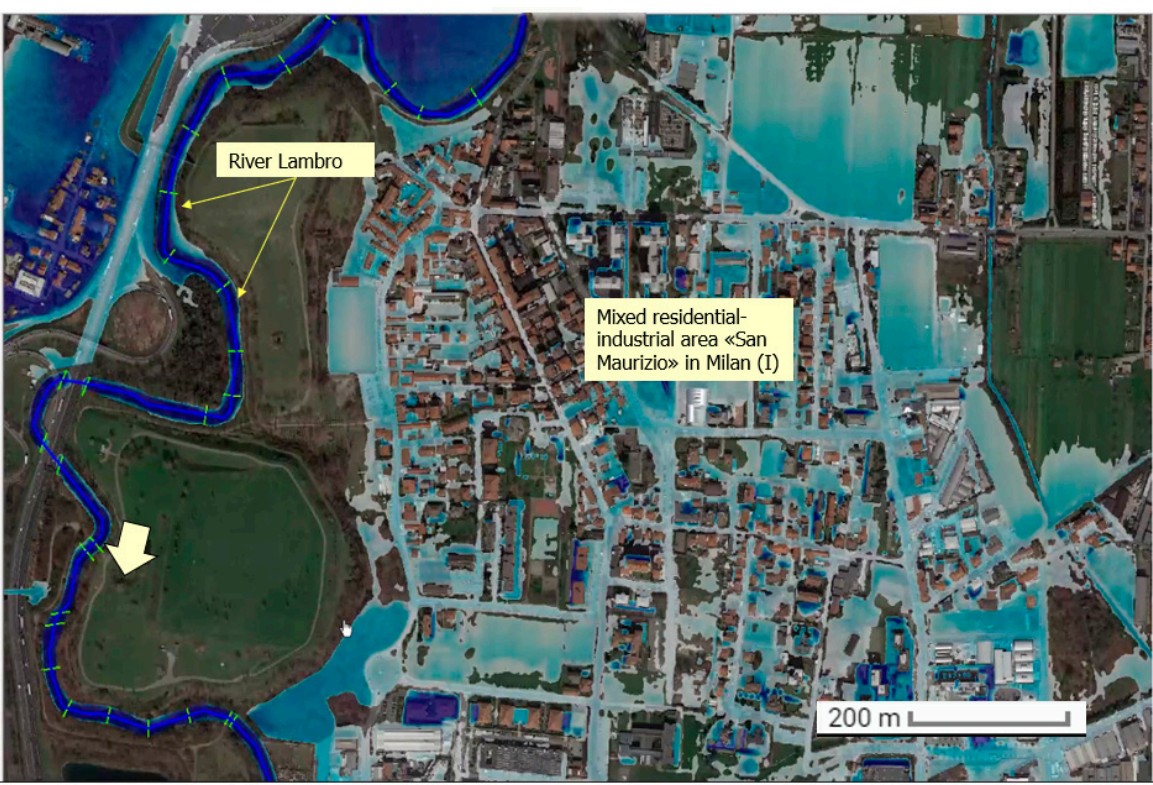

**Figure 7.** Rivers will flood urban areas as well; such areas must be equipped and prepared to minimize damages (Lambro river at San Maurizio al Lambro in Milan (I), in a simulation of a $T_R 200$ event, by kind concession of Alessandro Balbo).

Under the perspective of an extremely uncertain future, unavoidably characterized by a "bang-bang" climate (flood events alternating with drought events), it is perhaps obvious that "bang-bang" solution options, such as high levees, should be abandoned as they deliver incredibly dangerous systems either nullifying risk ("bang"), or failing with very big damages ("BANG").

In parallel, it would, however, not be wise to decide and put in place actions aiming to cope with the harshest possible scenario foreseeable now (strict min-max risk-averse criterion) because that would involve dramatic financial, social and political costs while the future climate may then prove to be milder. It is also important to consider that the cultural background will evolve progressively and so will social acceptance and political will. Nor we can ignore possible, future very bad scenarios by taking actions now that, in case things were to worsen, would prove counterproductive and incompatible with the needed changes. A significant flexibility should hence be incorporated into any plan, as stressed by the CRIDA approach (Climate Risk Informed Decision Analysis) recently promoted by UNESCO [57–59] (https://en.unesco.org/crida, accessed on 12 October 2021).

Here is therefore our new paradigm synthesized:

- Accept that safety does not exist, we need to *live and cope with risk* trying to minimize it, balancing costs, and adapting to futures now predictable only with very high uncertainty and no sound probability knowledge;
- Reduce risk by acting on all its components: *exposed value, vulnerability and hazard*. However, avoid as far as possible "grey" solutions because they imply an eternal, increasing economic burden on future generations (OMR costs), create a *fake safety* perception, while they increase real *residual risk* because they may (and do) fail. We rather need to reduce *fragility*, by limiting defenses to more modest works (that in case of collapse generate smaller, bearable damages), and more "green" infrastructures, i.e., *Nature Based Solutions*;
- Understand, respect and re-establish geomorphological dynamics (sediment transport and balance, space to wander, meander or avulse, longitudinal and lateral continuity) including the status, processes and role of riparian vegetation and woody debris;
- Design modular interventions with a flexible approach in order not to regret unnecessary costs borne now, nor to contradict present actions in the next future;
- Reduce residual risk ($R_R$) by equipping and preparing the territory, particularly outside the protected area, including urban zones. To this end:
    - reduce exposure (resettlement);
    - reduce vulnerability (real time alert systems, adaptation by *water proofing* and moving towards *hydro-cities*);
    - increase resilience (organization, insurance, etc. . . . ).

Here, *water proofing* includes all interventions aiming at reducing damages to buildings and infrastructures (e.g., provisional walls, sealing, anti-reflux valves; Figure 8) and the term *hydro-cities* denotes a city where in addition some streets are targeted and equipped to host water flows in case of bank or levees' overflows or collapse, and where some parks and squares are able to temporary store water—so as to alleviate the load on downstream areas—and where underearth crossings and basements are eliminated and/or very carefully managed and controlled.

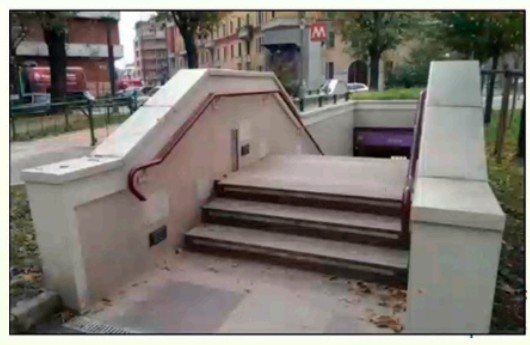
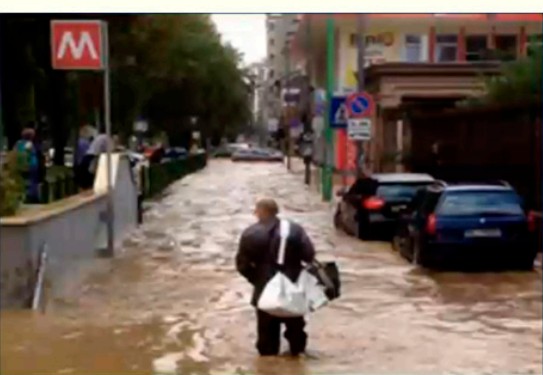

**Figure 8.** Example of water proofing intervention: (**left**) raised elevation of the contour of the access facility to a Metro MM5 station in Milan; (**right**) the same access proving to impede water entrance into the station, while keeping its full functionality, during a harsh flood from Seveso river in 2014 (by kind permission of Daniele Bignami, Politecnico di Milano, FLORIMAP project, Fondazione Cariplo, Milano, I).

The key idea underlying all this evidently is to preserve or restitute a wide fluvial space to the river, the *river corridor*, where the river can express as far as possible its River Style [60] with few inconveniences for humans. This may imply significant changes in the defense system as well as in the land use and water uses (withdrawals intakes, canals systems, dams, hydropower, navigation) as depicted in Figures 9 and 10.

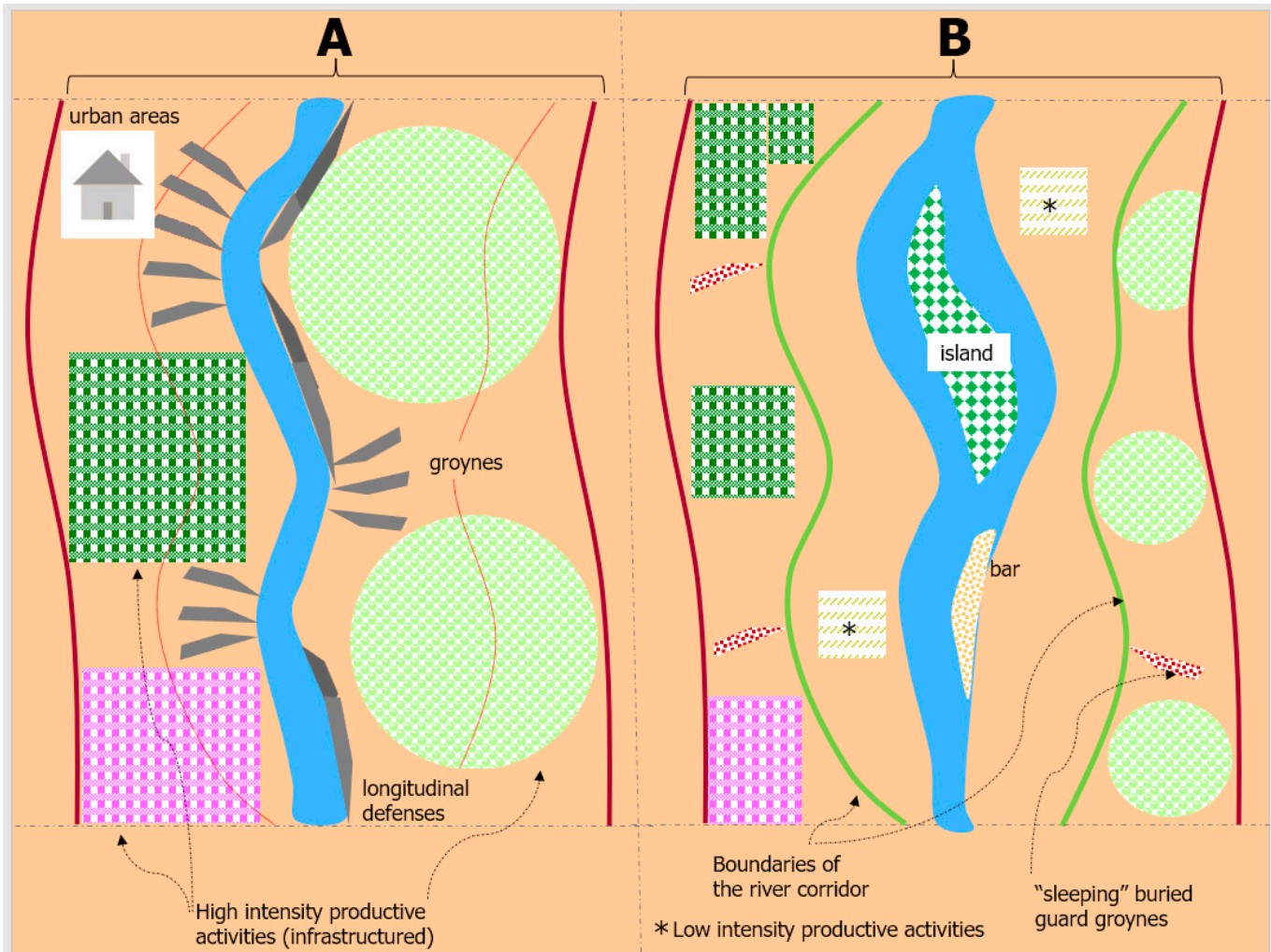

**Figure 9.** The paradigm shift: (**left**) typical present situation (**A**) of a river reach with heavy grey defenses, close-by urban areas, and the whole territory intensely utilized for agricultural and industrial purposes; (**right**) a wide river corridor (**B**) recreated in which the river can wander, meander or avulse so finding its own river style and recreating the suited geomorphic units (bars, islands, or oxbows, etc.), within a wide space delimited by clear boundaries (green lines; possibly not so external as the natural ones), defended by guard defense works (groynes generally buried or modest levees), coming into action only in case of an attempt to escape the boundaries; and where land uses have been modified, reducing the space occupied by those still inside the corridor and/or switching them to flood compatible types (indicated by *).

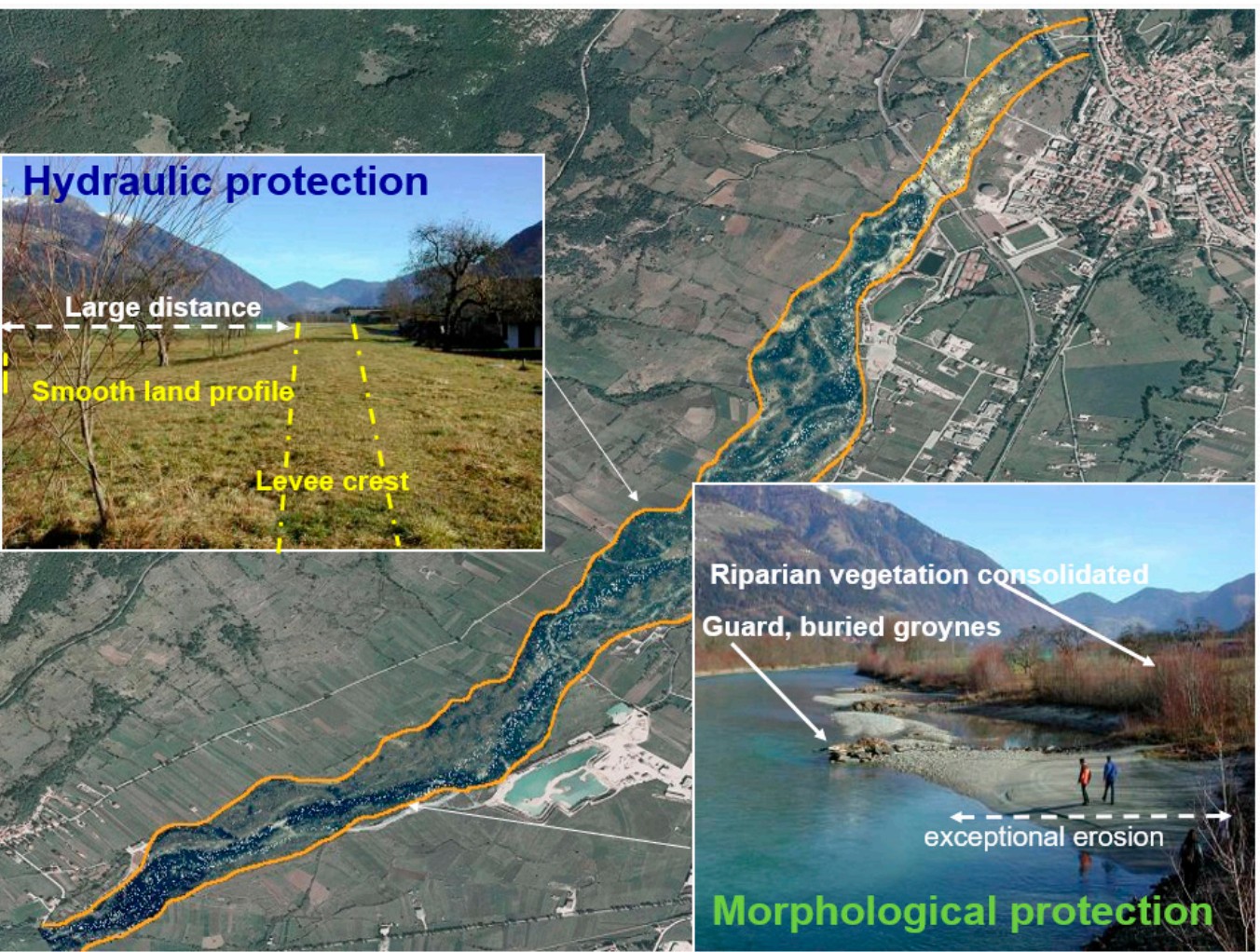

**Figure 10.** Hypothetical example of river corridor showing possible defense interventions to keep the river within the set boundaries (aerial photo of Sangro river at Castel di Sangro, Italy; pictures taken from Drava river project, by kind concession of Giuseppe Sansoni).

## 4. CBA as a Support Tool to Design River Corridors

It can be observed that the above claimed new approach is nothing else but what several guiding documents already state (e.g., [36,40,61–64]) and indeed what even the Flood Directive recommends. The progress I see is reassembling all the pieces into a set of concise statements and supporting them by a formalized framework (here below) that ensures some of the pieces are not forgotten, to look at their joint performance as a system, to ensure their coherence and to quantify verbal statements, so providing accountability and communicability.

CBA, on the other hand, can serve as a design tool, when comparing candidate ALTernatives of river setting. The "ALT" notation points out that an ALTernative is a complex figure involving engineering works, land use planning (the river space) and management mechanisms. An ALT refers here, ideally, to a whole river network in a basin or at least to a significant stretch of a river, not to a single, pointwise project. This is because the geomorphic and hydraulic behavior of a reach is, in general, very strongly linked to previous and successive reaches and as such should be dealt with through a system view. This consideration on spatial dependency does not prevent the scheme to be applied in real world problems, though; it is just a matter of introducing scenario components related to the boundaries where the system is cut: a section isolating an upstream or downstream reach; or the outlet in a lake or sea or into another river for which a boundary condition

is basically mandatory (typically water surface elevation). For simplicity, such pieces of scenario are herein considered as embedded within a climatic scenario.

In general, land use has to be changed within the river corridor, so its value V has now to be involved, as well as a transFormation cost F. Consequently, the approach of Figure 5 has to be widened by setting as objective function the following version:

$$
\begin{aligned}
&max\ [V(\mathbf{u}) - R_T(\mathbf{u}) - C(\mathbf{u}) - F(\mathbf{u})] \\
&\mathbf{u} \in \mathbf{U}^{***}
\end{aligned}
\tag{3}
$$

All terms are already defined above and differential with respect to a reference alternative (e.g., present situation where F and investment I are null); and the set $\mathbf{U}^{***}$ of decisions $\mathbf{u}$ now includes tools such as equalization schemes (i.e., administrative-financial mechanisms designed to promote land use change—with the progressive restitution of land to the river—with no economic impact on the stakeholders) and the Payment for Environmental Services (farmers can be paid to let their land be flooded sometimes, in order to protect downstream areas; e.g., [65] and for a historical perspective, see [66]). As already noted, additional components (e.g., externalities) can of course be added.

An operational scheme to apply CBA is shown in the following figure (Figure 11) that points out that the exercise is a bit more complicated by the need, in general, to consider that under a different climate—and because of a different size of the river corridor and a different set of defense and exploitation works—the river has a geomorphological response that will possibly lead it to a different morphology, geometry and behavior. This will go at high or (very) low speed depending on its nature (high or low energy, active or passive river), the availability of sediments and the frequency and intensity of flood events characterizing that climate scenario. Climate and the new morphology will determine a possibly different lifespan of works, which in turn contributes to determine their OMR cost.

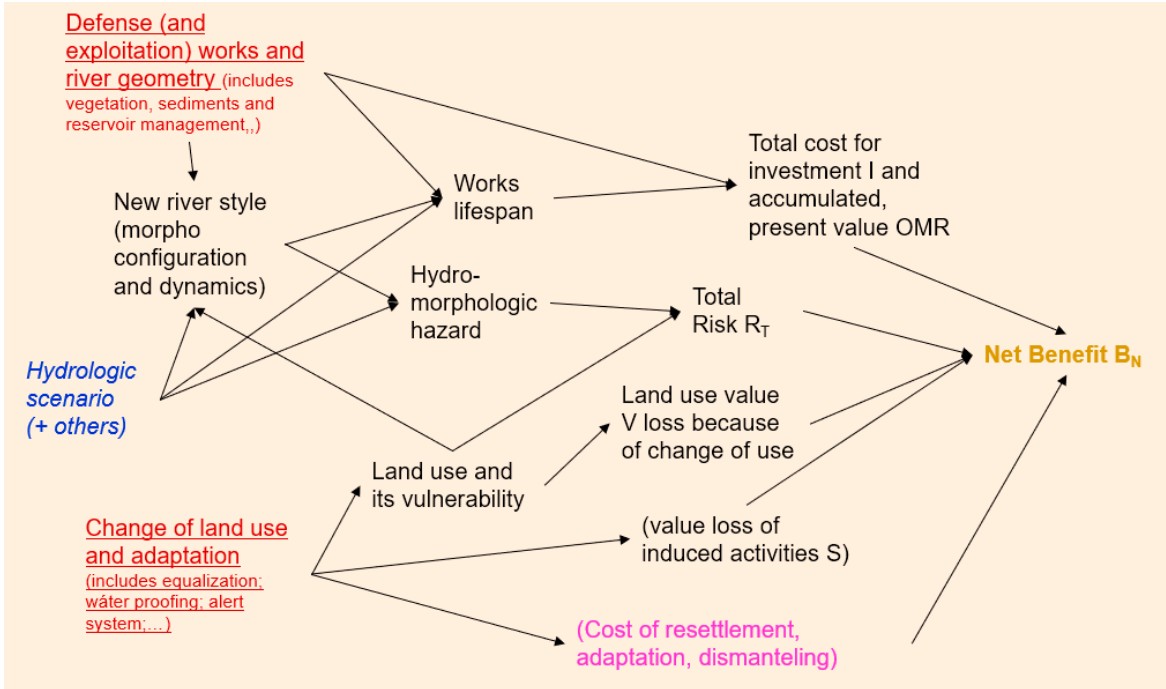

**Figure 11.** A scheme to carry out the CBA supporting the design exercise.

Notice that this scheme starts with a *hydrologic scenario*, i.e., a specification of the statistical characteristics of a water regime with its variability, corresponding to a given climate scenario.

In formulas, this simply says:

$$\Delta B_N = -\ \Delta R_T + \Delta V + \Delta S - \Delta I - \Delta OMR - F \tag{4}$$

where $\Delta$ denotes the differential with respect to a reference ALTernative ALT_0 (typically the "business as usual") and all other terms have been defined before in the text or in Figure 11.

This can be written in a more intuitive form so as to point out the meaning of designing a corridor, by indicating with $|x|$ the absolute value of x:

$$\Delta B_N = |\Delta OMR| - (\Delta I + \Delta R_T + |\Delta V| + |\Delta S| + F) \tag{5}$$

Here $\Delta OMR$ is assumed to be negative because in the corridor ALTernative (an NBS) there are typically fewer works and a much less aggressive river implying lower OMR costs, a significant economic saving. Few new works have to be built ($\Delta I > 0$) to protect the new corridor; the total risk is assumed to increase ($\Delta R_T > 0$), as defenses are reduced; land use value typically decreases ($\Delta V < 0$), behaving in the same way as costs, and so does the associated induced activities value $\Delta S$; while the transFormation cost F is, by definition, always positive. The overall figure $\Delta B_N$ may still be positive in a perhaps unusual case where in summary the economic saving of reduced defense works ($|\Delta OMR|$) justifies increased risk and/or loss of land use (this result was found, for instance, by [67]).

More common cases, as portraited in Figure 9, will rather see a decrease of land value (although equalization mechanisms can significantly reduce it or even cancel it) with a (significant) decrease of risk, a significant cost of transformation and some savings in terms of defense works (particularly in terms of OMR). It is also possible that an increase in the basic risk component R, is overcompensated by a decrease in the residual risk component $R_R$ because of more space assigned to the river, less fragile works (e.g., lower and farer levees), more adaptation, etc.

The important message of the CBA exercise, anyway, is that if $\Delta B_N$ is positive, then there potentially is a way to redistribute costs and benefits so that society as a whole is better-off. This means that, for instance, in case of increased risk, it is possible to compensate damages through the savings of works avoided and/or OMR costs spared. When $\Delta B_N$ is negative, it becomes key to ascertain the role of environmental services.

## 5. Dealing with Uncertainty

Uncertainty pervades the whole problem. Some key components can however be identified and managed. Figure 12 represents the main ones:

- *Climate*: this stems from the GCM modeling uncertainty (including incomplete knowledge of processes and insufficient or imprecise data to feed them) and the socio-political uncertainty about future climate altering emissions together with the natural unknown concerning volcanic or deep-sea emissions (which probably involve even greater figures). Therefore, it has both an epistemic component (linked to lack of knowledge) and an aleatory one (linked to physical processes and to political moves). Sea water levels can be seen as an element within this component. According again to Refsgaard, this can be considered as a scenario uncertainty if we restrict our world to the GCMs available; in any case, no probabilities are known. Hence, an ensemble modeling technique could be used for the GCMs' intrinsic variability; however, given the very uncertain emissions issue, a scenarios exploration technique seems more suitable;

- *Hydro-climatic variables*: given a hypothetical future climate, there still is a large aleatory uncertainty about the values assumed by climatic and hydrological variables at each time step of any year because of natural aleatory processes. According again to Refsgaard, this level of uncertainty can be assumed to be statistical, i.e., probabilities can be assigned, assuming that within a given climate, processes are ergodic. This component includes precipitation (p), temperature (t), wind speed etc.; yet, flowrates

Q are not included as they depend on the context and also on the existing river setting and the additional interventions foreseen by any ALT of flood management plan to be included. The hydro-climatic uncertainty can be dealt with by a classic statistical approach; details are provided below;

- *Context* represents a set of different factors potentially influencing the behavior of the system. With no claim of generality, I consider here a basic factor only, which is the possibility of collapse of works in a set of pre-defined sites of occurrence (e.g., historical failure sites, something assumed for instance in [17]), conditioned to the flowrate level Q in those river sections, with probabilities that can be estimated from historical events. According to Refsgaard, its level of uncertainty would hence be again statistical and a statistical approach can be suited for this case. Several other factors may be relevant, and can be considered, such as the functionality of defense works (e.g., a deviation canal may be occluded; mobile gates may be blocked or with no electric energy) or the downstream water level which acts as boundary condition when just a portion of a river basin is modeled (if the whole basin is considered down to the sea, its sea water level can be assumed to be associated with the climate scenario and hence known) and the degree of sediment filling of flood control reservoirs as well as their initial storage. Less technical issues may be relevant too, such as the forthcoming development of land use and associated infrastructures (new roads, new urban areas, etc.), or the prices of inputs required by the considered ALT (as discussed in particular in [27]). Concerning reservoirs and retention tanks, their water storage at the moment of a particular flood event is a consequence of the behavior and management of the system during the considered hydrological realization. As such, the water storage is, in principle, a known magnitude (given the initial storage at the beginning of the realization r) and not an *alea*; this is not true, however, if the simulation-management model that is adopted excludes (for the sake of simplicity) the reservoirs' management issue. Figure 12 also indicates in synthesis a possible way out to deal with such uncertainty components.

## 5.1. An Operational Scheme to Deal with Uncertainty

Operationally, this scheme can be translated into an algorithm along the following lines, trying to see at least a practical way to solve the posed problem:

For each climate k considered (obtained by a GCM model for a given future time), it is possible to generate a set of realizations $\mathbf{p}(k,r)$ with an appropriate time step t depending on the river basin considered (typically daily) and spatial resolution (s), where **bold** style denotes hereafter a vector. Techniques are available for this aim, for instance those shown in [68] or [69]. The number $n_R$ of the multi-year realizations must be large to capture the whole spectrum of possible events.

Given one (say ALT_x) of the candidate ALTernatives of river setting: For each one of the (spatial) Y-year long time series $\mathbf{p}(k,r)$ and of the context cases C(i/k,r) (e.g., a breach in one or more given sites of a certain levee), an integrated hydrological-hydraulic simulation is carried out and the corresponding flood field is obtained (depth h(s,t/i), velocity v(s,t/i), etc. in each site s and time t; the dependency on the particular ALT_x is dropped for simplicity herein. With that, the consequent damages $d_T(k,r/i)$ can be determined as a global figure integrated over the whole planning area and duration of the realization r (for simplicity we can think here of just direct, tangible damages; but they can be multidimensional as well). By construction, only the total damage is obtained, because this scheme already includes the residual risk associated with the occurrence of events exceeding the reference value for the defense works and their possible collapse. Of course, the value of exposed assets (from the land use and layout of infrastructures), their vulnerability as a function of the flood event characteristics (depth, speed, etc.) must be known for each category of exposed assets (an approach such as that of Paprotnya et al. [70] can serve here).

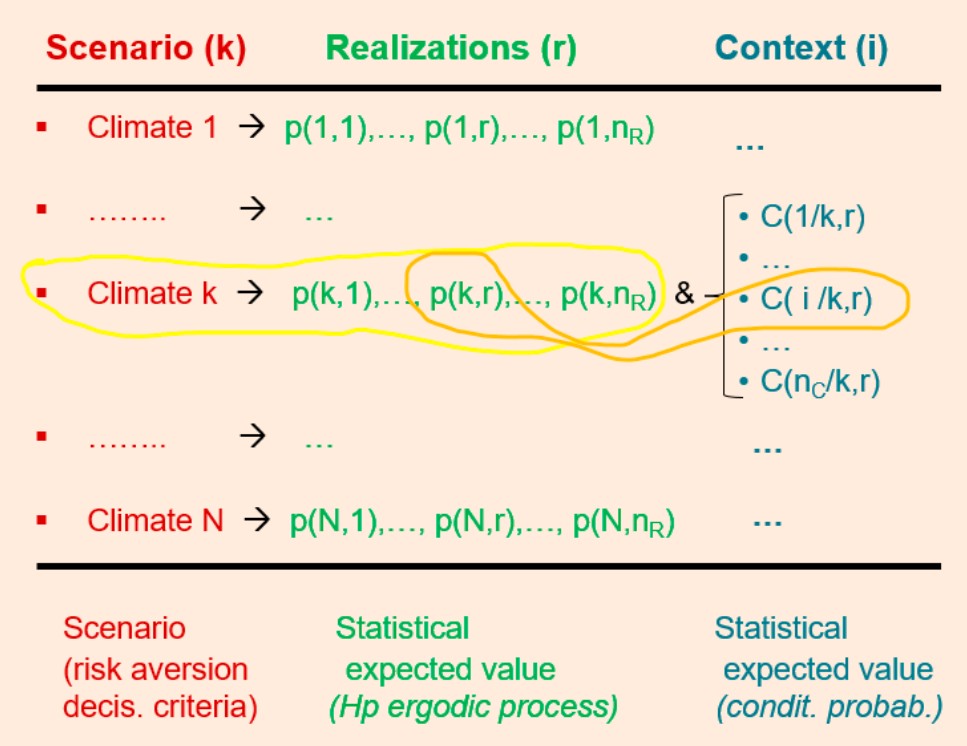

**Figure 12.** Representation of the uncertainty components. Given a climatic scenario, a set of $n_R$ long-term realizations (e.g., Y years) of corresponding climatic variables can be built (p, standing for spatio-temporal precipitation field, is indeed a vector of variables including temperature, wind speed, etc.); this is indicated by the light-yellow irregular oval for the generic climate k. Collaterally, any of $n_C$ context cases can occur (e.g., a levee breach in a given site) with a probability possibly dependent on the particular realization r of the kth scenario. The generic combination of events "realization r and context i" (out of the $n_C$ possible) is identified by the irregular, dark yellow loop.

Then, utilizing the probabilities of the context cases i (conditioned to the particular realization k,r), one obtains the expected damage by combining the damages corresponding to the same ALT x, but all possible context cases i:

$$d_T(k,r) = E_i\,[d_T(k,r/i)] \tag{6}$$

which is just a weighted average where conditional probabilities of context cases are the weights (the importance of considering the full array of magnitudes of flood events is well illustrated, for instance by Wobus et al., 2019 [71]).

Then, by recalling the "law of large numbers" (https://en.wikipedia.org/wiki/Law_of_large_numbers, accessed on 29 October 2021), thanks to the hypothesis of ergodicity of the climatological series $\mathbf{p}(k,r)$ within climate k, it is possible to estimate the annual risk as the expected value over the climatic realizations r as:

$$r_T(k) = sum_r\,[d_T(k,r)/Y]/n_R \tag{7}$$

where $n_R$ denotes the number of realizations r (each one Y years long) generated for climate k.

Finally, the global risk over the planning period T is obtained as the sum of the present value of the annual values, i.e.,:

$$R_T(k) = \sum\nolimits_{t=0,1,2,\ldots\,T}\,[\delta^t\,r_T(k)], \tag{8}$$

where $\delta$ is the discount factor.

This procedure can appear a bit of a "brute force" approach and indeed neither is it statistical, rigorously speaking. However, would it be worth to try to gain more rigor by extorting more information from scarce data and knowledge? An answer can be found in Box's sentence (although not concerning this specific problem), "... *it is inappropriate to be concerned about mice when there are tigers aboard*" (p. 792 [72]).

In any case, this procedure brings in an important added value, i.e., its ability to overcome the problem of selecting the magnitude of events of concurring tributaries of a basin which cannot be assumed small enough: the traditional schemes based on return periods simply cannot be applied.

### 5.2. Hints to Overcome the Computational Burden

This scheme is however evidently computationally heavy because it involves a high number of continuous hydrological simulations at a suitable, fine time step, over a (very) large area of a whole (sub)basin, plus coupled 1D-2D hydraulic simulations for each significant flood event where overbank flows may occur; namely, we have:

- $Nxn_R$ hydrological simulations, at daily time step along a Y-years long time series;
- $N \times n_R \times n_C \times n_f$ hydraulic 1D-2D simulations, where $n_f$ is the average number of significant flood events in each realization.

Particular care must therefore be put into the adoption of suitable models, at least until computer speed can be enhanced enough. In particular, the scheme is in general not tractable at present with physical-based models such as IBER (https://www.iberaula.es/56/iber-community/dissemination, accessed on 29 October 2021), or HEC2D (https://www.hec.usace.army.mil/confluence/rasdocs/r2dum/latest/introduction/hec-ras-2d-modeling-advantages-capabilities, accessed on 29 October 2021). On the other hand, a 1D model (e.g., HEC-RAS, https://www.hec.usace.army.mil/software/hec-ras/, accessed on 29 October 2021) cannot be used because it would not be able to deal with unexpected overbank and floodplain flows in large areas. An efficient modeling scheme can be set up, which integrates a quasi 2D model such as MODCEL [73–75] and a physically 2D model (say IBER or equivalent) or even a 3D model, MODCEL, and allows the rapid and reliable description of the flow within the main channels with a full 1D approach. It is also able to ensure the conservation of mass even when overbank flows occur, by modeling the exchange process and the dynamic flooding including effects on volumes and water elevation on large areas, modeled as a set of 0D cells with a quasi 2D effect; this solves the general simulation. A physically 2D model (say IBER or equivalent or even a 3D model) is then called into action in each case of works failure, or overtopping levees' thresholds, to reliably model the flood process in side areas where it is very dynamic and powerful, in order to determine the variables relevant for the estimation of damages where the velocity field, aside water depth, is an essential component. But a suitable coupling is required with the previous quasi 2D model as collapse abruptly changes the flow exchange law with the floodplain. This allows the proper computation of the damages accruing to the residual risk component.

### 5.3. Further Simplifications

A useful additional simplification is to consider at first just the different sub catchments involved, where no overbank flooding problems are envisaged, and carry out a simulation with a (semi)-distributed hydrological model (able to see in particular the different precipitation levels, as well as the river setting there defined by the considered ALT_x). The context case may influence the simulation fate (e.g., by provoking an overbank flow somewhere). The corresponding flowrates Q(s,t/k,r,i) of all sub catchments are hence obtained, which are then the inputs to the areas affected by flooding. Out of these time series it is possible to identify flood events (f) through criteria suitably defined (there may be more flood events f in a given year). Then, the heaviest simulation with the hydraulic model compound is carried out only for each one of such events with the scheme just

explained. Hence, a total damage $d_T(f;k,r)$ for each flood event f is now determined and the yearly expected risk would then be computed as:

$$r_T(k) = sum_r \{sum_f [d_T(f;k,r)]/Y\}/n_R \qquad (9)$$

The (delicate) climate-hydrological simulation may be even skipped when a simple basin with no tributaries and just a specific area with flooding problems is considered; in such cases, a simpler and more classic approach can be adopted to calculate risk. Indeed, as is usually executed in present practice, flowrate values $Q(T_R(j))$ associated with different historical return period $T_R(j)$, can be determined and incremented according to different climate change scenarios (k), together with a guessed shape of the associated hydrograms; with that, a 1D-2D hydraulic simulation is carried out; then, risk can be estimated as the summation over the different j of the corresponding damages weighed with the probability $\Delta P(j)$ computed from the same $T_R(j)$ (recalling that, by definition, $1 - 1/T_R(j)$ is the probability of not exceeding events of magnitude $Q(T_R(j))$, by assuming independence of events), that is:

$$\Delta P(j) = (1 - 1/T_R(j + 1)) - (1 - 1/T_R(j)). \qquad (10)$$

On the opposite side, a (very) challenging difficulty is the prediction of future morphology as, given a climate and an ALT_x, the river may change configuration and even style (e.g., [76]). Rigorously speaking, the whole time-evolving process should be considered; assuming the end-of-transition configuration (i.e., the new River Style) is however a possible simplification, although particularly drastic for low energy rivers.

*5.4. Decision Criteria under Strict Uncertainty*

In any case, once these calculations are carried out, when the ALTernatives are static plans, a decision matrix similar to that of Figure 13 is obtained. It is here that a decision criterion must be adopted. Several elaborated approaches have been proposed (a nice review is offered by [77], for instance the theory of imprecise probabilities [78]) dealing with sets of probability distributions rather than having to define one unique "true" distribution, and typically presenting lower and upper probabilities on an event or outcome of interest. Another alternative is the info-gap approach (e.g., [79]) where one starts with the best estimate of the future and then examines how ALTs perform as conditions depart increasingly from expectations, without identifying a worst case, and choosing the ALT that proves to be robust, i.e., still behaves satisfactorily, even at a very large horizon of uncertainty.

In a strict uncertainty context, the most straightforward and well-established risk-averse oriented approaches are however:

- *max-min*: avoid the worst performance, i.e., choose the ALT which behaves best under the worst scenario (e.g., [55]); or just any one which achieves an acceptable outcome under all scenarios (according to the criterion of [80]);
- *min regret*: try not to regret other choices, i.e., choose the ALT that would generate the minimum regret for not having chosen another solution, under the set of scenarios considered (e.g., [81]).

The emblematic situation of Figure 13 (right side) is here explained: (top) for each ALT, the green value indicates the best performance, while the pink one indicates the worst; (middle) under the max-min criterion (maximization), ALT 1 would be preferred because it is the one that provides the highest value under the worst scenario (90). Under the min-regret criterion (minimization), however, the best ALT would be ALT 3: its regret is 220, because under Scenario A, the lost benefit (regret) would be just (250 − 40 = 210) in case ALT2 had been chosen instead and (150 − 40 = 110) in the case of ALT1; while, under Scenario B, no regret would be experienced (because ALT 3 would prove superior); and, under Scenario C, the regret for not having chosen ALT 1 would be 120 and it is 220 for ALT2, which is its worst case (max regret); with a similar process, it can be verified that the other two ALTs would imply a higher regret.

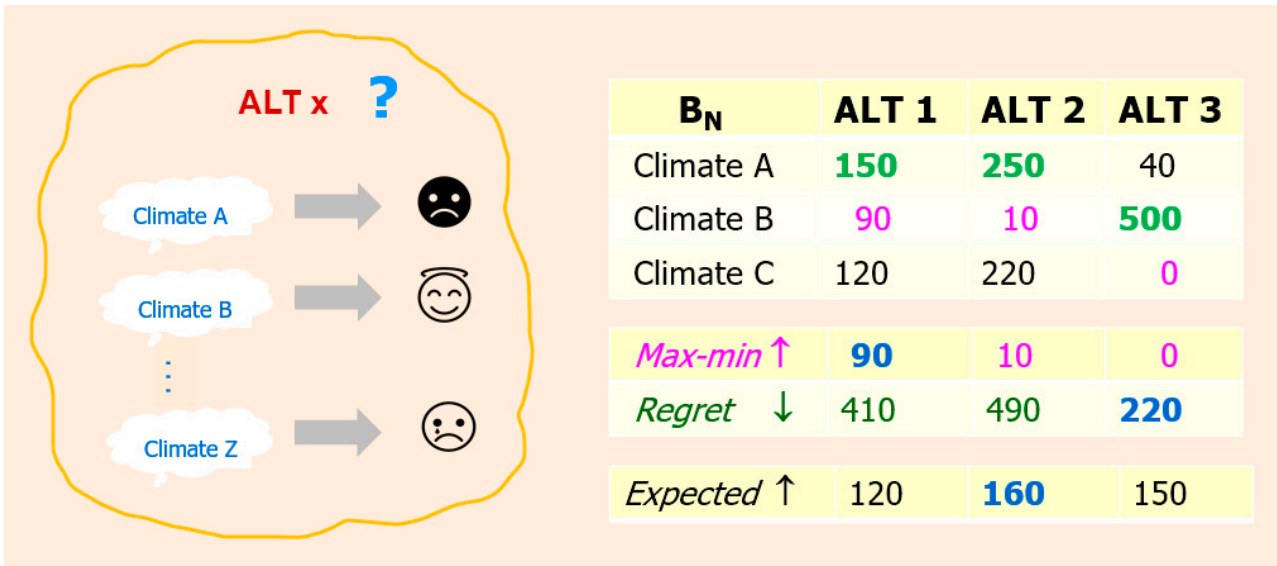

**Figure 13.** Deciding under uncertainty: different criteria are possible: (**left**) any ALTernative x may behave in different ways under different scenarios; (**right**) a simple case with three ALTs and three scenarios: according to the criterion adopted, the choice can be different (see text for explanations).

Finally, the result according to the classic expected value criterion is indicatively shown, just as an exemplification, in apparent contradiction with the initial statement that no probabilities can be assigned to the Scenario; here, the Scenarios are arbitrarily assumed equiprobable. It is interesting to see that each criterion leads to a different choice.

An additional complication comes from the intrinsic uncertainty of the hydrological and hydraulic models adopted, as well as from the description of the exposed value (land use, elements at risk) and its vulnerability. For the former, assuming a group of models can be set up, an ensemble approach would be a classic option together with a Monte Carlo approach (or more refined schemes such as the Latin HyperCube, e.g., [82,83]) on their key parameters; for the latter, a sensitivity analysis or, again, a Monte Carlo analysis can fit.

All this is to stress that uncertainty is deeply embedded in the problem formulation and solution approach, so that any claim of rigor in the analysis becomes questionable and a more qualitative interpretation of results is the sensible approach.

### 5.5. Flexibility Revisited

According to the new paradigm stated above, rather than a static plan, we should envisage a *strategy* where *flexibility* is an explicit characteristic because, as time goes by, interventions initially planned can be modified, anticipated or postponed. Things are hence a bit more complex (and much heavier computationally), but the problem can still be reconducted to the above reasoning as follows.

As climate is changing, at any time steps $\tau$ we can imagine to see a sequence of sets $K_\tau$ of climate scenarios (jointly with a corresponding end-of-period sea level). This is similar to discretizing the continuous time process of climate change. Given future courses-of-action, decision steps ($\tau$) occur when unbearable consequences would be experienced if nothing is changed (e.g., sea level reaches an unsafe threshold to be contained by a levee; or too frequent levee-overtopping events would occur), according to simulations.

Here below a possible formalization is introduced with the only purpose being to schematize the reasoning and render it in a concise fashion; in reality, its application may take a considerably different form because of socio-political and governance issues, but it still represents the technical essence.

At each decision step $\tau$, a finite (and reduced) set of possible decisions $\mathbf{u}_\tau$ is available which may be restricted in general by the courses-of-action $\mathbf{u}_0^{t-1}$ implemented until the last step $t(\tau) - 1$, i.e., because of the layout of the physical system and the legal-administrative

setting established insofar; this layout and legal-administrative setting is denoted here with $\mathbf{z}_\tau$ and is assumed to be governed by a deterministic dynamic equation of the type:

$$\mathbf{z}_{\tau+1} = \mathbf{L}(\tau, \mathbf{z}_\tau, \mathbf{u}_\tau), \tag{11}$$

where $\mathbf{L}(\bullet,\bullet,\bullet)$ is an appropriate transition function.

Figure 14 illustrates these concepts with a hypothetical example. In that figure, at the first decision step 0, the alternative courses-of-action (white dots) $a_\tau$ are: a levees-oriented choice (red), where Water proofing refers to interventions at the urban level to mitigate damages in case of levee overtopping or collapse; a hydraulic-oriented choice (blue), which is confident in the idea of controlling the event upstream as well as defending as far as possible the exposed assets until a further, costly protection level (Water proofing) is felt unavoidable; a time-taking choice (brown), before delicate transformations of the socio-economic organization is undertaken; two environmentally oriented choices (green), both starting immediately with a radical Nature Based Solutions plan including Land use change and Lower and Retreat levees and then differentiating with two alternative strong measures: either Raise the further levees and lower new levees, or carry out a Resettlement. This gives a set of six courses of action. Notice that raising levees is only possible after they have been reinforced; Lower and Retreat levees must go hand in hand with Land use change; and Resettlement does not make sense—or is at least very inefficient—once levees have been reinforced and raised; this exemplifies the idea that in general the course-of-actions $\mathbf{u}_0^{t-1}$ implemented until the last step $t(\tau) - 1$, may constrain the decisions set. More importantly, the implementation cost of future interventions may depend not only on the future decision $\mathbf{u}_\tau^{T-1}$, but also on the state $\mathbf{z}_\tau$, i.e., $C_{OMR} = C_{OMR}(\mathbf{u}; \mathbf{z}_\tau)$ (and so do the other components of the Net benefit expression of Equation (4)). For example, Lower and Retreat levees can be much less costly and more feasible if an appropriate Land use change had been previously carried out than in the case when it had not been; or in another case, rising levees would be much cheaper had they been built thinking of this future evolution as, otherwise, foundations would be certainly inadequate.

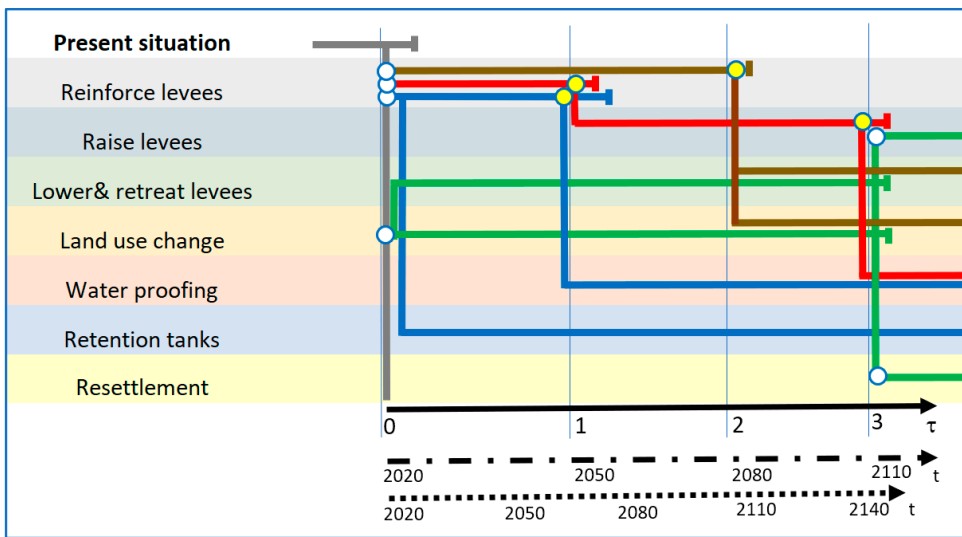

**Figure 14.** Some possible strategies for a river management problem (based on structural interventions) that can be undertaken as soon as current river setting is perceived to be inadequate (yellow dot) because of changed external conditions (particularly climate), denoted by a short vertical segment. This depends on the speed of climate change, as depicted by the lowest time axis (lower and faster speed, respectively) and determines decision steps at intervals of different duration. (Source: inspired by a similar figure from CRIDA UNESCO course 2021, https://openlearning.unesco.org/courses/; https://openlearning.unesco.org/courses/course-v1: CRIDA+CRIDA0001_ES+Run1_2021/about, accessed on 28 October 2021).

For this reason, the decision to be made at any time $\tau$ is not limited to a single action $\mathbf{u}_\tau$, it rather has to involve also all future decisions within the planning horizon T (a real option problem, e.g., [84]). In a simplified version, where one has to decide a course-of-action $a_\tau$ which specifies the future sequence $\mathbf{u}_\tau^{T-1}$ amongst the feasible ones belonging to a finite set $a_\tau = A(t(\tau), \mathbf{z}_\tau)$. The set $a_\tau$ can be built based on the general strategy, just by enumerating all the feasible and sensible combinations. As already noted, referring to Figure 14, at the initial step $\tau = 0$, this set includes six elements only. Of course, in a real case, things are much more complicated particularly because when a whole river setting is considered, decision options greatly multiply in terms of typologies, locations and possible combinations.

The problem addressed has the following structure conceptually (assuming that, to fix ideas, the *max-min* decision criterion has been adopted) and can be defined as a *strict uncertainty, finite horizon, open loop, optimal control* problem with a deterministic transition function $\mathbf{L}(\bullet,\bullet,\bullet)$:

$$\begin{matrix} max \\ a_{(\tau)} \in A(t(\tau), \mathbf{z}_\tau) \end{matrix} \left\{ \min K_{t(\tau)}, K_{t+1(\tau)}, \ldots, K_{T-1(\tau)} \left[ \Delta B_N(t(\tau), \mathbf{z}_\tau, a_\tau / \mathbf{I}_{t(\tau)}) \right] \right\} \tag{12}$$

$$\mathbf{z}_{\tau+1} = \mathbf{L}(\tau, \mathbf{z}_\tau, \mathbf{u}_\tau) \tag{13}$$

where the inner *min* operator applies to all the climatic scenarios sets $K_{t(\tau)}$ (each one analogous to that of Figure 12) from the decision step until the end T of the planning horizon, and $\mathbf{I}_{t(\tau)}$ is an information vector available at the real-world time $t(\tau)$ associated with the decision step $\tau$ and exploited in the construction of the set of climatic scenarios $K_{t(\tau)}$. The "open loop" connotation points out that no state information on the meteorological and socio-economic-political system generating the climatic uncertainty is considered explicitly, nor is the component $\mathbf{z}_\tau$ of the coupled layout state equation (Equation (12)), as that is a deterministic process completely defined given the past course of action $\mathbf{u}_0^{t(\tau)-1}$. The other component of uncertainty, hydro-climatic and context, are embedded, together with the hydrological and hydraulic system states' dynamics, in the definition of the objective function through the calculations presented in the previous paragraph and defined over a shorter time-scale.

Notice that in the previous generic formulation (Equation (4)) the argument $(\mathbf{z}_\tau, a_\tau)$ was implicitly embedded in the particular ALTernative considered, while the information vector $\mathbf{I}_{t(\tau)}$ was not made explicit; furthermore, the summation Equation (7) has now to be articulated to include the different future periods.

At each new decision time step $\tau$ a new information vector $\mathbf{I}_{t(\tau)}$ is available about the climate behavior and, accordingly, a similar problem is posed where the set of climatic scenarios is updated, as well as the layout state vector $z_\tau$ according to Equation (12).

The important thing is that, even if at each decision step one chooses a whole course-of-action, i.e., a kind of deterministic strategy, it is not said that that strategy will be adopted in full eventually, as at any decision time, if the whole strategy allows that, the process can jump over one of the different, feasible courses-of-action, either postponing an envisaged intervention or choosing to switch to a different one.

This type of scheme is a kind of adaptive management scheme known as Open Loop Feedback Control (OLFC), where *Feedback* refers to the updated knowledge of the climatic scenarios, and Open Loop points out that, at step $\tau$, no further information on the enlarged state of the controlled system (including the meteorological system generating climate) is assumed, for simplicity or ignorance, to be considered in the future decisions [85].

This problem, at any decision time $\tau$, can be addressed as a mathematical programming problem and solved by an exhaustive search on the possible $a_\tau \in A(t(\tau), \mathbf{z}_\tau)$, where, for each attempt of $a_\tau$ and for each climatic scenario k of the set defined for that period, risk is computed with a procedure analogous to the one presented above (ending with Equation (9)).

Alternatively, a classic backwards Dynamic Programming algorithm [85] can be adopted as the problem can be decomposed in a sequence of interrelated analogous, reduced problems, with the advantage to shorten the (heavy) simulations implied, particularly by the first steps where more $a_\tau$ have to be explored.

## 6. The Quality of Life, Multi Objective Paradigm

The CBA approach cannot be considered the last word for several reasons. Firstly, the most important part of the problem lies in the capacity of generating good candidate solution ALTs; this requires all the available scientific knowledge, including experience as well as the creativity, independently from any evaluation. Secondly, CBA suffers from severe conceptual and practical limitations that have been long since identified in the literature (e.g., [86–89]) starting from the indetermination of the discount factor; the incapacity to deal with non-monetary issues (e.g., the value of human life), here excluded for simplicity (they call for a multicriteria approach); and the inability to consider social issues as, for example, equity.

We can overcome this limitation by recognizing that what really counts to society is not just risk and costs, but eventually society's "quality of life" (QoL). With no claim at all to reduce psychological research (e.g., [90]) to a mere symbolism, it is sufficient to recognize that, amongst other components relevant to QoL, certainly we can find our friends $R_T$, C, as well as N, this latter being the value assigned to an ecologically healthy river, a very close relative of the ecological status of the Water Framework Directive (a thorough discussion on this element is provided in [91]). In other words, we decide to give to our problem a Multi Objective dress (e.g., [92,93]) that in its minimum form can take the following shape (Figure 15). Perhaps it is superfluous to note that win-win solutions are possible where all objectives are enhanced; but in most cases a trade-off must be searched for. Typically, N can be improved and R reduced, while C is increased. Actually, this balance is the essence of the challenge of harmonizing the Flood Directive with the WFD!

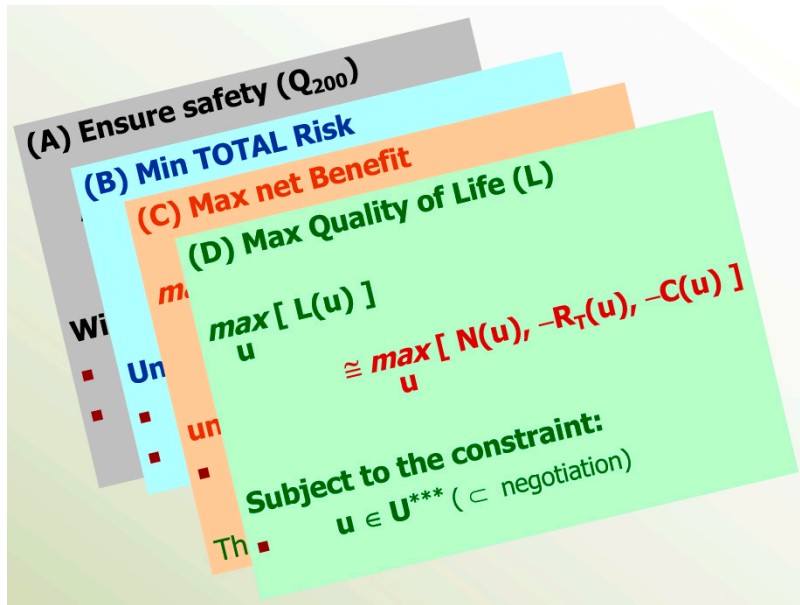

**Figure 15.** The problem that we, as a society, would actually have to approach where L stands for Quality of Life and the three objectives formulation (C: total costs; $R_T$: total risk; N: nature value) is a preliminary, partial, formulation.

This formulation perhaps hides the fact that the objectives are defined for the whole problem addressed, i.e., they necessarily are the product of a spatial aggregation (e.g., amongst different reaches of a river stretch; or those conforming to a given *water body*, according to the WFD definition; or those falling within a given country territory; see Figure 16).

This calls into play compensation schemes amongst different areas, typically between rural and urban areas, a kind of implicit solidarity. Nothing impedes, anyway, differentiating areas of particular significance, by splitting the objectives (and increasing the complexity). Even risk $R_T$, as already noted, can be split into several components of a multidimensional vector including non-tangible and indirect damages. In general, several of the fundamental objectives (defined in [91]) are to be called into play.

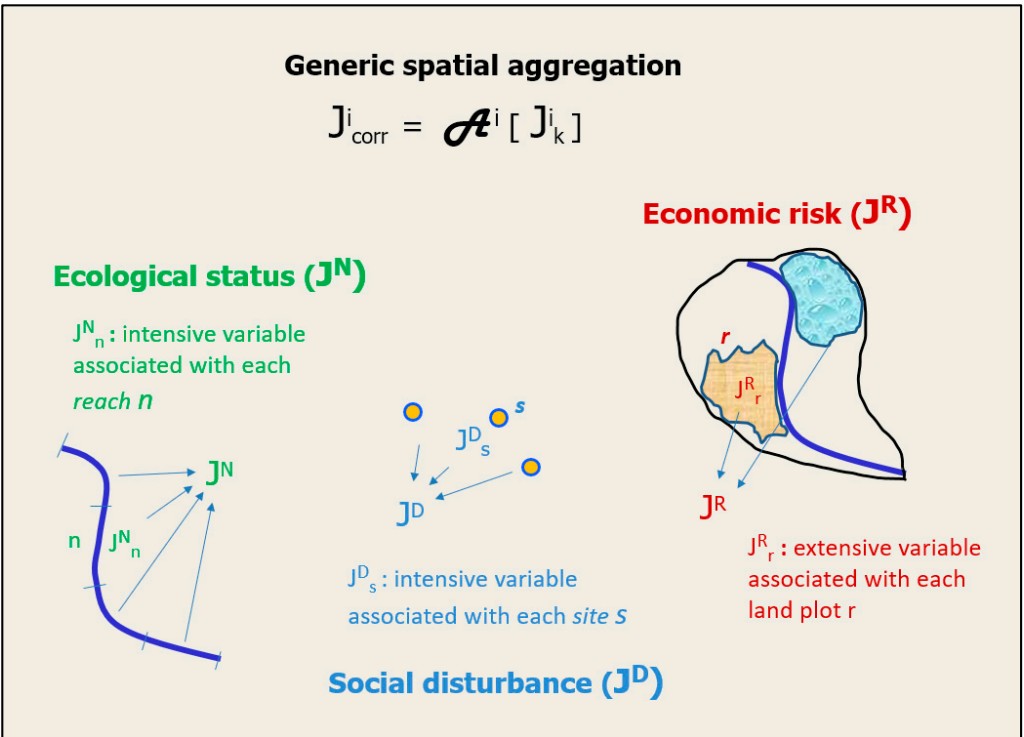

**Figure 16.** The generic i-th objective $J^i$ is the result of a spatial aggregation, according to a suitable operator $A^i$ [•]. Different spatial units ("pieces") may be involved: river reaches (k = n), in the case of the ecosystem quality objective N ≡ $J^N$ = $A^N$ [$J_n^N$]; several land units (land plots r: k = r) of a territory, in the case of the Risk objective R ≡ $J^R$ = $A^R$ [$J_r^R$]; some sites (k = s) in the case of the Disturbance objective D ≡ $J^D$ = $A^D$ [$J_s^D$] (e.g., hydropower generation loss in some plants; or loss of agricultural production in given properties). Depending on the formulation of the objective, the involved variables can be extensive (depending on the areas in the case of economic component of risk $J_r^R$; or the numerosity of affected hydropower plants), or intensive (e.g., ecosystem quality, in the case of $J_n^N$).

### 6.1. Incorporating the Multiobjective Scheme within a Participatory Decision Making Process

An articulated decision process that would be focusing on this kind of approach to plan the new river corridors and their associated river settings requires a high level of socio-cultural-political maturity. Indeed, the multi-objective approach must find an appropriate place on the Decision Makers' table, a fact that, in general, cannot be taken for granted, although notable exceptions do exist, such as the "Room for the Rivers" policy in the Netherlands [94] which develops an interesting analysis on how relevant effects have been achieved. In addition, as our problem may involve considerable land use changes and may be influenced by several policies, the ability to predict the future evolution of a complex system is required and to this end a mix of several approaches can be adopted, from mechanistic modeling, to qualitative nexus analysis or back-casting, i.e., the exploration of multiple path-ways that can lead to a desired goal (e.g., [95]). Indeed, the framework proposed in this paper focuses on the generation of meaningful, far-sighted river setting alternatives, inspired by the river restoration approach (e.g., www.ecrr.org,

accessed on 20 September 2021) and on the evaluation of their performance, but not on the indispensable social process that should support and feed the whole process.

To this end, participation of involved stakeholders and of the general public is a key component that has to come into play from the very beginning to identify and understand the problem, providing indispensable pieces of information, as well as to generate promising solution options and ALTernatives, relying on experience and debate. A characteristic component of it is the insurgence of interest or social conflicts. To manage this reality, the whole process depicted so far should be developed within a participatory framework (as, by the way, the WFD and FD ask for) and this could rely on three evaluation levels (Figure 17):

(i)     the *Technical level*, i.e., the multi-objective evaluation which allows to see with clarity what is the level expected to be reached by each ALT, so discarding those clearly inefficient or unacceptable for some reason; the quantification of objectives is intended to overcome the weaknesses already identified by many (e.g., p. 390 [96]) concerning the rare definition in explicit terms of collateral purposes of flood control planning, such as improving spatial quality;

(ii)    the *Quality of Life* level, where the satisfaction of each interest group feeling affected is evaluated (possibly including the risk components disregarded so far), interest conflicts are addressed and where negotiation actually takes place leading to new ALTs, ideally through a creative, constructive, iterative process;

(iii)   the *Strategic level*, where the general welfare of society is considered by the illuminated, non-corrupted Decision Makers that we all wish for. This latter would take into account a global judgment on a) the *quality of life* and b) another on the intra- and *inter-generational fairness* of the decision. The former would collate the satisfaction expressed by stakeholders, together with a consideration of the rest of society through the output of an extended CBA exercise which reflects the efficiency in the allocation of resources (indicating, theoretically, a good satisfaction of needs and no waste; e.g., [87]); furthermore, it would include a judgment on the actual feasibility of the ALT, as well as on the ability to maintain it (financial sustainability), for, without them, no QoL can be reached and/or sustained. The intra- and inter-generational fairness would consider an appropriate distribution of pros and cons amongst social groups and the conservation of environmental assets for future generations [89]. It would therefore explicitly deal with one of the advocated weaknesses regarding equity issues, as raised [27] in front of the US Army Corps Principles and Guidelines of 1983 [26].

*6.2. Governance: An Indispensable Pillar*

This evaluation structure should be immersed in an adaptive decision process where decisions are periodically reconsidered (as actually the FD and WFD require) not only because new information on the climate becomes available and new, more reliable forecasts can be issued (which feeds the OLFC scheme presented above), but also because new projects come into play and society evolves, so awareness and preferences change.

All this articulated structure cannot come into being if it is not institutionalized, which calls for a sufficient level of governance. This is a progressive conquest, as pointed out with clarity for instance by [97,98], referring in particular to the lower Mississippi, shows how water management regimes have changed over time and that major transitions were preceded by niches, in which new visions were developed and empowered.

However, in spite of its importance, this topic is not discussed here as there are valuable references covering this issue (e.g., [99–101], the latter discussing in particular governance challenges and identifying mechanisms that proved effective). The framework here presented is just a natural complement, towards a more rational decision-making process.

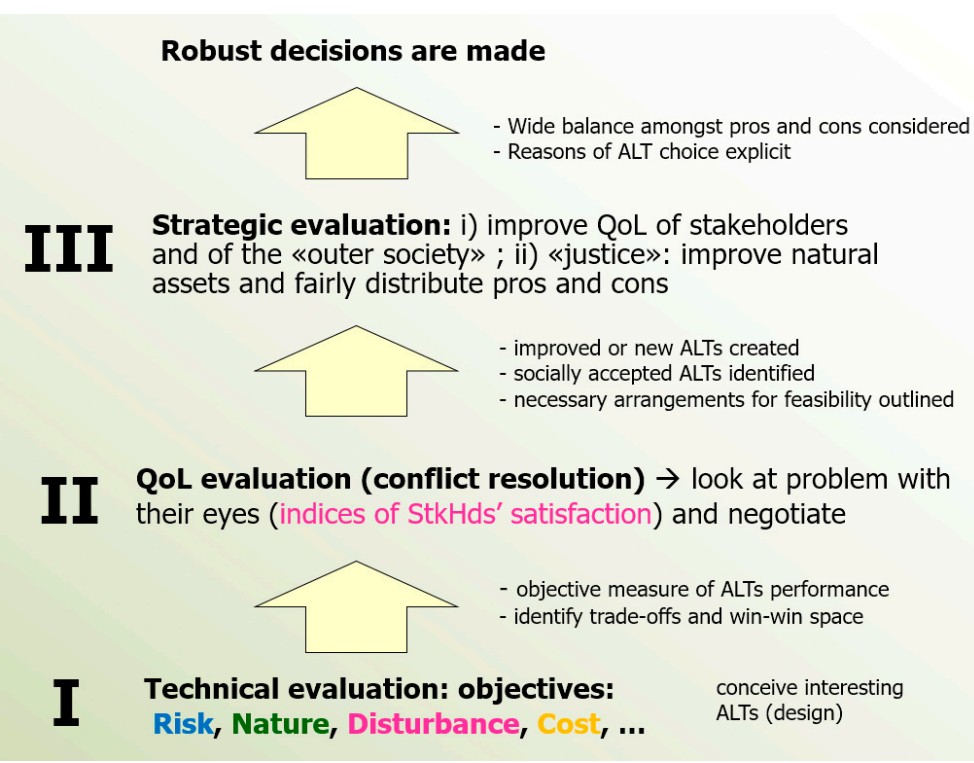

**Figure 17.** A three-level evaluation framework to support a thorough participatory process.

## 7. Conclusions

### 7.1. Weaknesses of Current Practice of Flood Management Planning

Flood Risk Plans required by the EU Flood Directive are in general not sufficiently far reaching, often filled with declared principles and innovative criteria fully shareable and even daring, consistent with the discussion here presented, but then in practice they keep a too tight adherence with the current setting, born long ago under the classic engineering paradigm of "putting the territory in safe conditions" (this is the case in particular of the Italian plans). Climate change is being considered as an anomaly, rather than a constitutive forcing factor of future land use and river setting. Planning is still undertaken under a time horizon definitely too short to allow for serious changes of river setting and land use, with scarce consideration for sea levels rising and serious modifications of the climatic processes. Geomorphological risk is generally not addressed. A unified, consistent, modelling approach and toolbox is lacking. The stated measures do not really try to shift the current situation towards a deeply renovated and more suited configuration that would include socially, economically and politically daring changes. Cost Benefit Analysis is mostly seen as a tool to prioritize interventions decided with qualitative criteria and procedures. The measure of stated objectives eventually refers only to the action-lines envisaged (i.e., an intermediate level), not to the fundamental objectives, and when win-win solutions are claimed, no clear measurement of the expected performance and tradeoffs is provided. The effects of new economic mechanisms to foster changes (insurance, PES, taxation) are not looked for in concrete terms at the evaluation level and are seen as integral components of solutions, as indeed they are. Strategic Environmental Assessment is too often an additional burden that further complicates the management of real problems, rather than being a structural part of the planning process (i.e., something embedded within the three level evaluation of Figure 17). Some of these considerations are already remarked on in [102], although referring to the previous version of the FRMPs, but very probably the next version will still show similar weaknesses.

*7.2. Contribution of This Paper*

This paper provides an articulated reasoning aiming at convincing researchers, stakeholders, public servants, decision makers and the general public that we should dare to undertake more challenging actions to combat flood risk (and gain other services) in view of a very worrying future. The core of such actions is to guarantee more space to the rivers (*wide river corridors*) either by avoiding new land consumption or by restituting space occupied by settlements, infrastructures and other uses. A deep adaptation is also required of the anthropogenic tissue in order to "gently live with flood risk", both within the rural and the urban environments. This adaptation requires us to physically modify our widely diffused defense infrastructures (levees, longitudinal bank protections, weirs, etc.) in order to recover a much higher dose of the natural geomorphic dynamics of rivers, as well as to modify buildings, roads and the urban setting as a whole so that less damages would occur in case of an event, while a higher contribution to flood retention and/or to flood conveyance would be ensured (*hydro-cities*). Legal, administrative, financial, socio-cultural, formal and informal mechanisms are required to achieve a sufficient level of governance, without which the required transition would be impossible.

This paper also provides a conceptual-operational framework to transform current, mainly qualitative, planning approaches into a quantifiable process. It intends to synergistically use, on the one side, Cost Benefit Analysis as a support to screen and design possible Alternatives of new River Settings (including land use within the river corridor), and, on the other side, an articulated Multi Objective, Multi Criteria approach, as a support for a negotiation process where people's wellbeing and environmental conservation are at the center.

The proposed scheme visualizes a way to manage the inevitable, extremely high uncertainty involved, including a formal, operational problem structuring for a progressive adaptation of policies.

It can also help to develop constructive dialogues with stakeholders as it provides intuitive pieces of information (the quantified objectives) that would definitely provide a key information to stakeholders. As such, it can hopefully stimulate a fruitful debate towards an improved practice.

The reasoning is synthesized in Figure 18 and is schematically referenced here:

- the present setting of rivers and their basins is unsatisfactory in large part because of the classic engineering approach to hydro-morphological risk management and its widespread socio-cultural acceptance;
- in the future climate change will make this situation harsher;
- a significant improvement is achieved by addressing explicitly the residual risk, both hydraulic and morphological;
- this awareness, together with the consideration of the importance of economic aspects and the strict link between residual risk and Operation, Management and Replacement costs (OMR), leads to an improved strategy to define river setting based on a number of principles (Section 3.3) as well as on recognizing that uncertainty towards the future is enormous and we are basically ignorant about future probabilities;
- an evolved approach to river management stems from this strategy, supported by Cost Benefit Analysis and a suitable modeling framework able to consider changes in the hydro-meteorological behavior of river basins and to determine risk (including residual risk) in an integrated, straightforward manner;
- meaningful, efficient ALTernatives (and flexible strategies) of river setting and fluvial corridor are so generated that can feed a thorough participatory decision process supported by a three-level evaluation scheme;
- the evolution of society (and its values, perceptions, etc.), and the ever-updating knowledge of the actual climate, call for an adaptive scheme that can lead us to better rivers and improved Quality of Life (QoL);
- all this must be institutionalized and supported by the two pillars of governance and participation, within a democratic framework, of course.

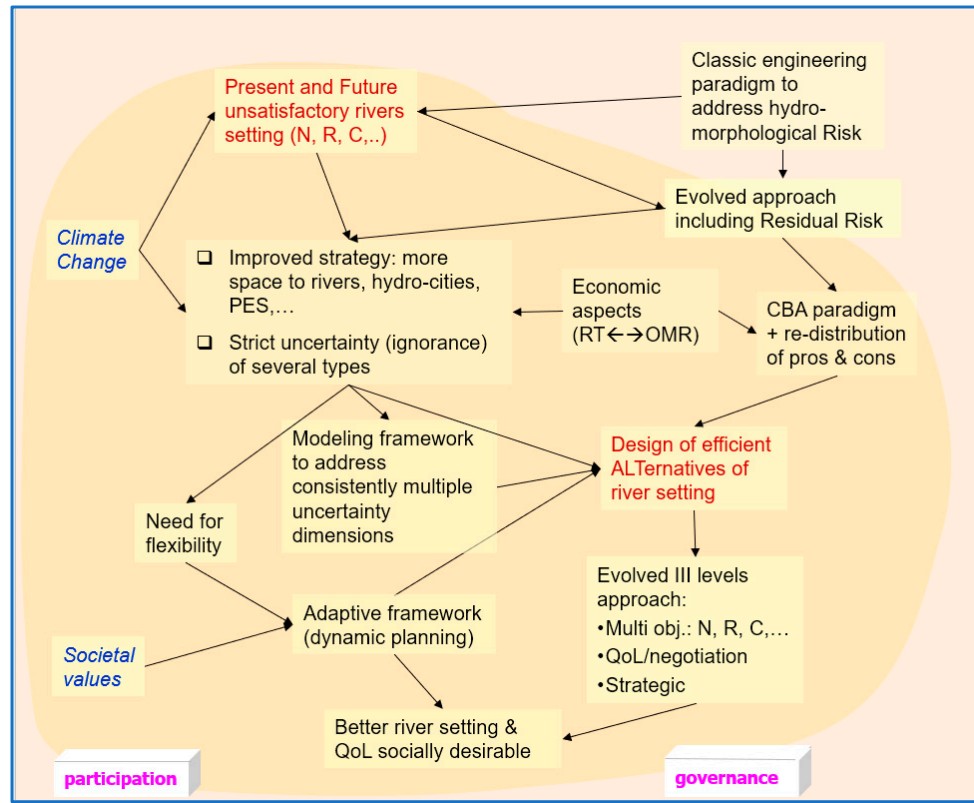

**Figure 18.** The reasoning developed in this paper (see text for explanation).

### 7.3. Improving Decision Makers Agenda

Understandably, the priority of Decision Makers is, generally speaking, to comply with legal obligations; this means—in Europe—first of all producing in time the due versions of the Flood Risk Management Plans and their components. However, the reasoning here developed calls for a kind of political-social earthquake. As events will be harsher, and transformation of the anthropogenic system requires a significant time, actions should be undertaken from now, incorporating a series of issues:

○ assume a much longer planning horizon than is currently envisaged, something of the order of a century or more in order to decide actions now, while looking at the far future, while ensuring sufficient flexibility to adapt decisions afterwards;

○ overcome historical time series and switch to climatic predictions, defining consistently a set of the most credible climate scenarios (including sea level rise), and their evolution, specific to each river basin;

○ develop a powerful dissemination-education campaign and establish permanent communication channels, while building or reinforcing the governance capability of society;

○ establish trans-disciplinary working groups: land use planning, water resources management, risk control, nature conservation, environmental assessment;

○ set up verified, operational methods/tools to address the required technical steps: transforming precipitation into flowrates; simulating the hydraulics within the river and the territory, including the possible collapse of works; predict geomorphic change of rivers; assess exposed value and vulnerability for different risk components; build comprehensive evaluation indices (one for each fundamental objective), ensuring internal consistency;

○ identify, through the working groups, strengths and weaknesses of the territory and of the river system (by analyzing the historical behavior and current status) and start delineating very different Alternatives of river corridor in syntony with the diverging points of view and aspirations of stakeholders; these Alternatives should

incorporate explicitly new economic-financial-administrative mechanisms to foster changes (insurance, PES, taxation, social agreements, etc.) amongst measures;

○ in this process, adopt the basic cost-benefit analysis not just to prioritize options, but as a design tool of Alternatives;

○ undertake an open, pragmatic and honest participatory process where the different objectives are quantified first, and the pros and cons assessed afterwards, and in which candidate Alternatives are progressively modified and improved until a decision is to be made;

○ document the whole process in a synthetic form, easy to be periodically updated and easy to be consulted (different to what is currently produced, where kilometers of papers have to be read each time and information is dispersed).

It may seem that an approach such as the one proposed here is too far looking, mechanistic and somehow prepotent. However, why should we not be able to put it in practice if our cousins, the Romans, dared to plan the birth of new cities (e.g., Nîmes in France) by first looking at the sites where abundant and clean water sources were available, then studying the aqueduct path (with an extremely careful eye on slopes and elevations) and so eventually establishing the site at which the city could be born, to be served by gravity with extraordinary works that lasted centuries?

**Funding:** This research received no external funding.

**Institutional Review Board Statement:** Not applicable.

**Informed Consent Statement:** Not applicable.

**Conflicts of Interest:** The authors declare no conflict of interest.

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
