# Peer review of "Making Room for Our Forthcoming Rivers"

_water, doi:10.3390/w14081220_

Round 1
Reviewer 1 Report
The paper has significantly improved. As a final comment, please adapt the conclusion section in a way that is more than a sysnthesis. I would like to the author to propose an agenda for policy makers, based on the synthesis.
For example, the bullet on "in the future climate change will make this situation harsher" -> what does that mean for policy makers?
Author Response
Answer to Rev 1
Dear Reviewer 1, thank you again for you time and constructive suggestions. I think I better grasped now your thought and hence reworked once more the conclusions . I hope now they satisfy your expectations.
Kind regards

Reviewer 2 Report
Thank you for responding to the peer-reviewed opinions.
I have confirmed that the author responded to all the indications.
Author Response
Answer to Rev 2
Thanks a lot. Kind regards.
Reviewer 3 Report
It is a good study and worth for publication. I feel like the current format is more like a technical report rather than a scientific paper. Although it is a concept paper. It would be better if the author can prepare the figure or the structure well.
A revision is necessary.
Author Response
Answer to Rev 3
Dear Reviewer 3, I am not sure I understood your thought. Anyway, I slightly revised the figure synthesizing the logical structure of the reasoning (Fig.18) and I reviewed once more the whole paper trying to improve readability. I believe that a concept paper deserves a language and style quite different from that of a typical scientific paper. I hope anyway this version can satisfy you. Kind regards.
Reviewer 4 Report
Thank you for submitting your paper to water. I read carefully manuscript number: water-1660588, the manuscript entitled: "Making room for our forthcoming rivers". Check the English Grammar. The English language is poor. Please check all parts of the manuscript and correct grammatical errors. The authors should ask the help of native English speaking proofreader, because there are some linguistic mistakes that should be fixed. Nevertheless, the manuscript is not acceptable in its current form. Since the quality of the study is lower than average for the publication in the journal, tools for objective function optimization are unclear in the methodology and conclusion, but it still needs a major revision before reconsideration. I attached my reviewer comments in the PDF file. Authors should apply all of my comments.

Author Response
Dear Reviewer, thank you for the time you dedicated to comment the paper. In the attached PDF you will find my reply to each one of your comments, not always in agreement, though. English mistakes (at least most of them) have been corrected. That the quality of the study is lower than average comes, I guess, from a misinterpretation of the nature of this paper: this is not a “scientific” paper filled with data taken by rigorous methods, but a straightforward, logical reasoning, based on up-to-date scientific knowledge. This to me, does not mean at all that it is worth less. On the contrary, I firmly believe that it is only through an open minded reflection on current empirical and theoretical basis that policy improvements can be achieved.

This manuscript is a resubmission of an earlier submission. The following is a list of the peer review reports and author responses from that submission.
Round 1
Reviewer 1 Report
Thanks for addressing my review comments from which the paper has improved significantly. Before the paper can be accepted for publication, I recommend the author to consider the following minor concerns I have with the paper:
- Please include a separate section with information on the research design
- Please be aware that ref #98 concerns the Mississippi River, whereas Ref #97 is generic.
- The conclusion section should be improved by carving out the insights presented earlier in the paper e.g., what are novelties or burning policy questions for the future?
- I find the English writing style ok, but not per se academic. E.g. lines 984-985: why using the word shy? The paper includes numerious examples of strange wording.
Reviewer 2 Report
The authors point out the applicability of their analysis to the EU. It would be a bit different for the US. This is OK to me as the limits of the paper are well defined.
Reviewer 3 Report
Dear Author,
I confirm my impression: this kind of work cannot be published in a scientific journal, as it reports your personal point of view without evidence. This is not a concept paper, but just an opinion paper.
Reviewer 4 Report
The author addressed some of my earlier comments, which helped improve clarity and flow of information to a certain extent in the revised version.
However, the manuscript still needs extensive “major revisions” that would require re-writing of many sections and very careful editorial reviews throughout. The manuscript still has unnecessary, confusing, and distractive verbage, unclear, unsubstantiated, confusing, and misleading statements, in addition to lengthy and confusing compound sentences, improper word choices, and many grammar mistakes.
The topic is interesting but writing style is way below standard, which completely obscures the main message, focus, and contribution of this work. The author should use plain language and proper technical terms to get his points across. I needed to read some compound sentences several times and still I have no idea what the author wants to convey with those sentences.
In addition, in several places, the authors resorted to over-exaggerated statements. Such misleading and over-exaggerated statements need to be removed.
In my opinion, the discussion in line # 265-281 is right on target and exemplary in terms of proper writing style and information provided. This writing style and format should be considered as ‘template’ and applied to the entire manuscript.
Here are some examples in support of my critiques above:
- Line # 16: A spontaneous question arises: “new climate -> new river”?
why ‘->’ need to be used in this sentence? Can this not be expressed in a simple and clear sentence something like : How climate change impact river systems?
More importantly, the author does not seem to be interested in formation of ‘new’ rivers due to changing climate in this manuscript, but seems to be interested in hydrological alterations of river systems under a changing climate. This is one of the key points in this manuscript and should be explained very clearly and accurately in the beginning of the manuscript.
- Line # 65: “because of our ignorance on future probabilities distributions of hydrological variables”.
Please avoid such overgeneralized and over-exaggerated sentences. Please be specific about who is really ignoring probabilistic analysis? Probabilistic analyses have been key to all the engineering projects I have been involved and we’ve never ignored them.
- Line # 111. “Ensure that for a reference event (e.g. a 200 years return period flowrate Q200) the associated 111 flood risk R200(u) is nullified”. Is this a hypothetical statement? If so, please clarify it in the beginning of the sentence. Otherwise, this is an incorrect, impractical, and unrealistic statement. How can risk be nullified? Such risk can ONLY be alleviated/ reduced/minimized etc. by developing and implementing proper mitigation and adaptation strategies but cannot be nullified.
- Line # 165: “Rs and R” Distinction between Rs and R is not clear. In line #160, risk associated with flow exceeding the design flow capacity of a mitigation structure was defined as Rs. Similarly, R was defined in line 167-168 as the risk associated with a mitigation structure placed in to protect an area of interest from flooding. So, what is the difference between R and Rs?
- Line # 121. “Damages occur because rivers are not “clean”. “clean” was used here incorrectly even it was written in “ “. The word ‘clean’ refers to water quality in a river or sediment quality in streambed. The term ‘clean’ is related to ‘pollution’ or ‘contamination’ of a river system. On the other hand, the author apparently tried to refer to “reduced flow capacity/ability of the river” by the term ‘clean’. It is very confusing. It can simply be stated as “flood damages could occur due to reduced flow capacity in the river channel as result of accumulation of sediments in the riverbed and/or invasive vegetation. Flow in the river could, however, be restored by dredging of sediments and removal of unwanted vegetation” or something like that. Complete and clear sentences with plain language and proper technical terms would enhance the clarity and readability.
- Line # 173, “because we now admit that risk cannot be eliminated, but just reduced”. This is a known fact and applicable to all engineering structures, as none of the engineering projects has endless time and budget for operation and maintenance.
- Figure 2. “Before” and “After”. Apparently, the author does not consider the potential role and use of spillways in the dam design. If this is the case, this assumption (although impractical and unrealistic) needs be mentioned in the figure caption with proper justification.
- Line # 104-115: The arrow sign in math is commonly used to represent the cause ->result’ relation. It is completely opposite here. All these expressions can all be written in plain and clear sentences. For example, “keep the water within the channel, avoiding overflows -> levees” can be written using concise, clear, and plain language, such as “risk for a river overflowing its banks can be reduced by building levees” Or something like that.
- Line # 113. “structural and non structural interventions”. Structural mitigation systems (e.g., levees) are clear from the discussion. How about non-structural interventions? This term appeared with no explanation in the text.
- Line # 123 (also in 125, 133, 145 …): “Defense work”. It is called ‘mitigation strategies/structures/plans’ and have been implemented to reduce flood risk.
- Line 134: “at the critical moment because occurring in an unexpected period”. Grammatically incorrect unclear sentence.
- Line 149: that of a socially and environmentally (and economically) very valuable asset. Why “and economically” is in parenthesis here?
- Line 157: “although with a lower probability”. Please be clear with “probability” of what? Probability of overflooding a dam structure?
- Line 160: “an almost zero figure, As the territory at risk usually has its main portion outside’. Unclear sentence.
- Line # 184. “cost of waiving the compactness of CBA.” what is the compactness of CBA?
- Line # 39: “extremely large and enormous” doses. Why both extremely large and enormous in this sentence?
- Line # 54: ‘organic and robust discourse” what does really “organic” discourse mean?
- Line # 109: “planning approach that can be given a formal shape”. What does “formal shape” mean here?
- Line #114: “exposed value”. What is exposed value?
- Line # 114: “grey-ones”. “Mitigation structures” (to reduce flood risk) is a well-adapted technical term in flood risk and protection analyses, which would be clear to the reader, unlike ‘grey-ones’. What is the purpose of using such unclear terminologies over well-established terminologies?
- Line # 118: “the collective imaginary that”. What is collective imaginary? This compound sentence is unclear. It needs to be re-written.
- Line# 186-192: A very long, unclear compound sentence. Please break it into a few sentences and write them in plain language.
- Line #112-115. Various symbol, c(u) with boldfaced u or a scalar u. What do scalar and boldfaced u represent? The author should clearly describe each variable/symbol when they appear the first time in the text.
- Figures 3 and 5. They are unnecessary. The relevant equations and brief explanations should be embedded into the text. Moreover, In Operation Research, ‘subject to’ is used instead of ‘under constraint’ in formulating optimization models.
- Line # 200: “to a quite unpleasant, but unavoidable guest of”. Unnecessary verbage.
- Line #228-231. Unnecessarily complicated and confusing compound sentence. It needs to be simplified.
- Line # 329. “ we are literally ignorant about the future as any probability estimate”. I disagree with this sentence and use of the term ‘ignorant’. Indeed, we’re not ignorant. We have future predictions - based on our current understandings - sometimes with high uncertainties, but we’re continuously learning and making progress.
- Line 332-337. Discussion on ‘kunk’ is unnecessary and should be removed.
I have stopped my comments here. This manuscript needs substantial editorial and technical revisions before it is considered for a technical review. In its current version, it is unsuitable for a technical review.
Reviewer 5 Report
Title : Making room for our forthcoming rivers
[Overview]
Overall, the authors present an interesting scheme and concept for making flood control actions.
I think that the current content is sufficient, but there are several points that I felt should be corrected, so please check the comments below.
[Minor comments]
Whole paper : The resolution of some of the introduced figures (Fig.2,4,6 and 8) is low. Please replace with a clear image.
p.5 : line 169
*** zero figure, As *** Isn’t this “A” lowercase?
p.8 : Fig.6
This figure is taken from Reference 47, but when using the figure itself in this paper, please explain so that the reader can understand the contents of the figure. The meaning of the red plot is explained in the caption, but I think it's better to explain what the other plots mean.
Fig.14,16,18
I felt that the explanation in the captions of these figures was too long. If you need a long explanation, please write it in the text, not in the caption.
